# Riemannian Stochastic Optimization for Sufficient Dimension Reduction

Thibault Pautrel [* 1]   François Portier [* 2]

## Abstract

Sufficient dimension reduction (SDR) makes high-dimensional regression tractable by projecting the covariates onto a low-dimensional subspace that preserves the conditional mean of the response. Existing gradient-based estimators either operate in the ambient space and suffer from the curse of dimensionality, or localize in the reduced space at a per-outer-iteration cost at least quadratic in the sample size. We show that minimizers of the population Minimum Average Variance Estimation (MAVE) risk approximate the same Grassmannian target as the Outer Product of Gradients (OPG), and recast the empirical criterion as a smooth maximization on the Stiefel manifold with closed-form Riemannian gradient. The resulting algorithm, SMAVE, combines sparse projected-space nearest-neighbor localization with Riemannian stochastic gradient ascent. A simplified version comes with almost-sure convergence and a non-asymptotic rate matching the standard non-convex stochastic first-order scaling. Empirically, SMAVE matches or improves on RMAVE's synthetic subspace recovery at moderate-to-high ambient dimension, and on four real datasets it uniformly improves over OPG and is competitive with or outperforms RMAVE at orders of magnitude lower runtime.

## 1. Introduction

Sufficient dimension reduction (SDR) seeks a low-dimensional linear projection $\mathbf{B}_*^\top X$ of high-dimensional covariates $X \in \mathbb{R}^p$ that preserves all information relevant for predicting a response $Y$. Unlike unsupervised techniques such as principal component analysis, SDR exploits the supervised structure: the projection captures label-

relevant variation rather than overall data variance. When such a projection exists with $d \ll p$, the intractable problem of estimating the regression function $\mathbb{E}[Y \mid X]$ in $\mathbb{R}^p$ reduces to the tractable problem of estimating $\mathbb{E}[Y \mid \mathbf{B}_*^\top X]$ in $\mathbb{R}^d$. Beyond computational savings, this projection provides an interpretable, low-dimensional representation of the covariates for the supervised learning problem.

This goal places SDR within a broader family of supervised representation learning methods. Metric learning provides a concrete example: learning a Mahalanobis distance $d_M(x, x') = \|x - x'\|_M$ adapted to prediction tasks (Ghojogh et al., 2022; Kulis et al., 2013) reduces to learning a linear projection when the metric matrix $M$ has low rank. If $\mathrm{rank}(M) = d \ll p$, then $M = \mathbf{B}\mathbf{B}^\top$ for some $\mathbf{B} \in \mathbb{R}^{p \times d}$, and $d_M(x, x') = \|\mathbf{B}^\top(x - x')\|_2$, which is precisely the Euclidean distance in the projected space that SDR methods seek to identify.

Two broad families of methods have emerged for estimating the subspace spanned by $\mathbf{B}_*$ from $n$ independent samples, each navigating a fundamental tension between statistical flexibility and computational cost. Inverse regression approaches (including Sliced Inverse Regression (SIR; Li, 1991), Sliced Average Variance Estimation (SAVE; Cook & Weisberg, 1991), and Directional Regression (DR; Li & Wang, 2007)) exploit moment conditions on $\mathbb{E}[X \mid Y]$ to achieve $\sqrt{n}$-consistency with low computational cost, but require linearity or constant variance conditions that often fail in practice. Gradient-based methods offer greater flexibility by estimating the subspace directly from local derivative information, avoiding these distributional assumptions. The Average Derivative Estimator (Härdle & Stoker, 1989) and Outer Product of Gradients (OPG; Xia et al., 2002) construct subspace estimates from pointwise gradient estimates, while Minimum Average Variance Estimation (MAVE; Xia et al., 2002) formulates the problem as local polynomial regression. However, this flexibility comes at a computational price: the kernel bandwidth scale causes neighborhoods to become exponentially sparse when $p$ is large. Refined variants like RMAVE (Xia et al., 2002) and structure-adaptive OPG (Hristache et al., 2001; Dalalyan et al., 2008) address this by localizing in the projected space $\mathbb{R}^d$, but still incur a cost at least quadratic in $n$ per refinement step from dense pairwise kernel weights. A separate line of work, initiated by Fukumizu et al. (2009),

[*]Equal contribution  [1]Laboratoire des Signaux et Systèmes (L2S), CentraleSupélec, Université Paris-Saclay, Gif-sur-Yvette, France [2]ENSAI, CREST, Bruz, France. Correspondence to: Thibault Pautrel <thibault.pautrel@centralesupelec.fr>.

*Proceedings of the 43ʳᵈ International Conference on Machine Learning*, Seoul, South Korea. PMLR 306, 2026. Copyright 2026 by the author(s).

characterizes dimension reduction subspaces through conditional independence in reproducing kernel Hilbert spaces (Wu et al., 2019; Park et al., 2023), offering a distinct perspective that we do not pursue here.

Since the target is a subspace rather than a specific matrix, the natural parameter space is the Grassmann manifold $\mathrm{Gr}(p, d)$ of $d$-dimensional linear subspaces of $\mathbb{R}^p$. For computation, we represent each subspace by a matrix with orthonormal columns, optimizing over the Stiefel manifold $\mathrm{St}(p, d) = \{\mathbf{B} \in \mathbb{R}^{p \times d} : \mathbf{B}^\top \mathbf{B} = \mathbf{I}_d\}$. Riemannian optimization on these matrix manifolds is well-established in numerical linear algebra (Edelman et al., 1998; Absil et al., 2008) and increasingly adopted in machine learning, from decentralized training with orthogonality constraints (Chen et al., 2021) to parameter-efficient fine-tuning of large language models (Park et al., 2025). We bring these tools to sufficient dimension reduction, demonstrating that scalable Riemannian stochastic gradient methods can unlock both statistical and computational gains for foundational problems in nonparametric statistics.

We propose **SMAVE** (Stochastic MAVE), a scalable algorithm that combines Riemannian optimization with adaptive localization in the projected space to achieve competitive or sharper subspace recovery than RMAVE at a fraction of the computational cost.

**Contributions.**

1. **Theoretical characterization.** We establish that MAVE minimizers approximate the span of the regression function's gradients (the same Grassmannian target as OPG) but recovered through projected-space local regression rather than ambient-space gradient aggregation (Propositions 2.4–2.5). This unifies two major gradient-based SDR families under a common target, and reveals that the MAVE objective is intrinsically a function on the Grassmannian, motivating a Riemannian treatment.

2. **Riemannian reformulation.** Building on the above, we reformulate MAVE as maximizing a smooth, $\mathcal{O}(d)$-invariant objective on $\mathrm{St}(p, d)$ (Proposition 2.7) and derive its Riemannian gradient in closed form (Proposition 4.1).

3. **Scalable algorithm with convergence guarantees.** SMAVE combines mini-batch Riemannian stochastic gradient steps with sparse $k$-NN weights in the projected space, periodically refreshed as the subspace estimate evolves. This reduces per-iteration cost from the quadratic-in-$n$ scaling of RMAVE's outer iterations to a linear-in-$n$ scaling of SMAVE. For a simplified version, we prove almost-sure convergence to a critical point (Proposition 4.3) and an $O(1/\sqrt{mT})$

non-asymptotic rate (Theorem 4.4), matching the optimal rate for non-convex stochastic first-order methods.

4. **Empirical validation.** On synthetic benchmarks, SMAVE reduces subspace estimation error over RMAVE while running one to two orders of magnitude faster, with the gap widening as the ambient dimension grows. On four real regression datasets spanning $p \in [11, 128]$, SMAVE uniformly outperforms OPG, is broadly competitive with RMAVE at low and moderate $p$ at a fraction of its runtime, and significantly outperforms RMAVE at $p = 128$.

**Organization.** Section 2 formalizes the SDR problem and introduces our local approximation framework. Section 3 reviews related work, including gradient-based SDR methods and Riemannian optimization. Section 4 presents the SMAVE algorithm and establishes its convergence guarantees. Section 5 reports experiments on synthetic and real data. Proofs and additional results appear in the appendix.

## 2. From Sufficient Dimension Reduction to Local Regression

### 2.1. Sufficient Dimension Reduction

Let $(Y, X)$ be a random vector with $Y \in \mathbb{R}$ and $X \in \mathbb{R}^p$, and let $g(x) := \mathbb{E}[Y \mid X = x]$ denote the regression function. When $p$ is large, estimating $g$ directly suffers from the curse of dimensionality. The goal of sufficient dimension reduction is to find a matrix $\mathbf{B}_* \in \mathbb{R}^{p \times d}$ with $d \ll p$ such that

$$g(X) = \mathbb{E}[Y \mid \mathbf{B}_*^\top X]. \tag{1}$$

When such a projection exists, the intractable problem of estimating $g$ in $\mathbb{R}^p$ reduces to estimation in the low-dimensional space $\mathbb{R}^d$.

*Remark* 2.1 (Central mean subspace). Condition (1) defines the *central mean subspace*, weaker than the classical *central subspace* requiring $Y \perp X \mid \mathbf{B}_*^\top X$. The central mean subspace suffices for regression and is the natural target of gradient-based methods (Xia et al., 2002).

Condition (1) exhibits a key invariance: it holds for $\mathbf{B}_*$ if and only if it holds for $\mathbf{B}_* \mathbf{A}$, for any invertible $\mathbf{A} \in \mathrm{GL}_d(\mathbb{R})$, since $\mathbf{B}_*^\top X$ and $\mathbf{A}^\top \mathbf{B}_*^\top X$ generate the same $\sigma$-algebra. The natural parameter space is therefore the Grassmann manifold

$$\mathrm{Gr}(p, d) = \{\mathrm{span}(\mathbf{B}) : \mathbf{B} \in \mathbb{R}^{p \times d}, \ \mathrm{rank}(\mathbf{B}) = d\},$$

consisting of all $d$-dimensional linear subspaces of $\mathbb{R}^p$.

For computation, we represent each subspace by a matrix with orthonormal columns. The Stiefel manifold $\mathrm{St}(p, d)$

comprises all $p \times d$ matrices satisfying $\mathbf{B}^\top \mathbf{B} = \mathbf{I}_d$. This representation is not unique: $\mathbf{B}, \mathbf{B}' \in \mathrm{St}(p,d)$ span the same subspace if and only if $\mathbf{B}' = \mathbf{BQ}$ for some $\mathbf{Q} \in \mathcal{O}(d)$, yielding the quotient identification $\mathrm{Gr}(p,d) \cong \mathrm{St}(p,d)/\mathcal{O}(d)$. Optimizing over $\mathrm{St}(p,d)$ simplifies implementation, provided the objective is $\mathcal{O}(d)$-invariant, a property we verify below. The Riemannian optimization machinery (tangent spaces, gradients, and retractions on the Stiefel manifold) is detailed in Appendix A.

The natural approach to finding $\mathbf{B}_*$ is to minimize the prediction error on the Stiefel manifold:

$$\min_{\mathbf{B} \in \mathrm{St}(p,d)} \left\{ D(\mathbf{B}) := \mathbb{E}[(Y - \mathbb{E}[Y \mid \mathbf{B}^\top X])^2] \right\}.$$

Since $D(\mathbf{B}) = \sigma^2 + \mathbb{E}[(g(X) - \mathbb{E}[Y \mid \mathbf{B}^\top X])^2]$, where $\sigma^2 = \mathbb{E}[\mathrm{Var}(Y \mid X)]$, minimizing $D(\mathbf{B})$ amounts to approximating $g(X)$ by the projected regression function $\mathbb{E}[Y \mid \mathbf{B}^\top X]$ in $L_2(\mathbb{P})$.

## 2.2. Local Approximation Analysis

Direct optimization is intractable since $\mathbf{B}$ appears inside the conditioning event. The MAVE approach (Xia et al., 2002) introduces a local surrogate: for $x \in \mathbb{R}^p$ and $u > 0$, define

$$D_{(x,u)}(\mathbf{B}) := \min_{a \in \mathbb{R}, \, \mathbf{b} \in \mathbb{R}^d} \mathbb{E}_{(x,u)}\left[ (Y - a - \mathbf{b}^\top \mathbf{B}^\top (X - x))^2 \right], \tag{2}$$

where $\mathbb{E}_{(x,u)}[\cdot]$ denotes expectation conditional on $X \in B(x,u) = \{ y \in \mathbb{R}^p : \|y - x\| \leq u \}$. When $u$ is small, the optimal $a$ approximates $g(x)$ and $\mathbf{Bb}$ approximates the projection of $\nabla g(x)$ onto $\mathrm{span}(\mathbf{B})$.

Let $P_\mathbf{B} = \mathbf{BB}^\top$ and $G_{(x,u)} = \mathrm{Cov}_{(x,u)}(X)$ denote the orthogonal projection and local covariance, respectively.

**Assumption 2.2** (Smoothness). *There exists $L > 0, u_0 > 0$ such that, for $\mathbb{P}$-almost every $x$ and all $u \in (0, u_0]$,*

$$\mathbb{E}_{(x,u)}\left[ |g(X) - g(x) - \nabla g(x)^\top (X - x)|^2 \right] \leq \frac{L^2}{4} u^4. \tag{3}$$

**Assumption 2.3** (Local covariance lower bound). *There exists $\lambda \in (0,1], u_0 > 0$ such that, for $\mathbb{P}$-almost every $x$ and all $u \in (0, u_0]$,*

$$G_{(x,u)} \succeq \lambda u^2 I_p. \tag{4}$$

Assumption 2.2 is satisfied whenever $g$ has bounded second derivatives, while Assumption 2.3 holds whenever $X$ admits a density bounded away from zero on its support; the constraint $\lambda \leq 1$ is necessary and discussed in Appendix B.

**Proposition 2.4** (Local scale separation). *Under Assumptions 2.2–2.3, for all $u \in (0, u_0]$:*

*(i) If $u \leq \frac{\lambda}{2L} \|(I_p - P_\mathbf{B})\nabla g(x)\|_2$, then*

$$D_{(x,u)}(\mathbf{B}) \geq \mathbb{E}_{(x,u)}[\sigma^2(X)] + \frac{\lambda u^2}{2} \|(I_p - P_\mathbf{B})\nabla g(x)\|_2^2.$$

*(ii) If $\nabla g(x) \in \mathrm{span}(\mathbf{B})$, then*

$$D_{(x,u)}(\mathbf{B}) \leq \mathbb{E}_{(x,u)}[\sigma^2(X)] + \frac{L^2 u^4}{4}.$$

The proposition establishes a *scale separation*: subspaces containing $\nabla g(x)$ incur excess risk of order $u^4$, whereas subspaces with $\|(I_p - P_\mathbf{B})\nabla g(x)\|_2 > 0$ incur excess risk of order $u^2$ as $u \to 0$. Integrating over $x$ yields a global characterization. Define $D_u(\mathbf{B}) := \int D_{(x,u)}(\mathbf{B})\, \mathbb{P}(\mathrm{d}x)$ and the gradient outer product matrix $\mathcal{M} := \mathbb{E}[\nabla g(X) \nabla g(X)^\top]$.

**Proposition 2.5** (Global gradient alignment). *Under Assumptions 2.2–2.3, any minimizer $\tilde{\mathbf{B}}$ of $D_u$ satisfies*

$$E(\tilde{\mathbf{B}}, \mathcal{M}) := \mathrm{tr}\left( (I_p - P_{\tilde{\mathbf{B}}})\mathcal{M} \right) \leq \frac{9L^2}{2\lambda^2} u^2.$$

**Geometric interpretation.** The discrepancy $E(\mathbf{B}, \mathcal{M})$ measures the $\mathcal{M}$-weighted projection error on the Grassmannian $\mathrm{Gr}(p,d)$ of $d$-dimensional subspaces of $\mathbb{R}^p$. Writing $\mathcal{M} = \sum_i \mu_i v_i v_i^\top$ in its eigendecomposition, we have $E(\mathbf{B}, \mathcal{M}) = \sum_i \mu_i \sin^2 \theta_i$, where $\theta_i$ is the angle between $v_i$ and $\mathrm{span}(\mathbf{B})$. Minimizing $E(\mathbf{B}, \mathcal{M})$ thus seeks subspaces aligned with the dominant eigenvectors of $\mathcal{M}$. The matrix $\mathcal{M}$ is precisely the population objective of OPG; Proposition 2.5 shows that MAVE targets the same subspace (i.e. the top eigenspace of $\mathcal{M}$) through local regression rather than gradient aggregation.

*Remark* 2.6. The localization can be relaxed from balls to cylinders $B_\mathbf{A}(x,u) = \{ y : \|\mathbf{A}^\top (y - x)\| \leq u \}$ for any $\mathbf{A}$ with $\mathrm{span}(\mathbf{B}) \subseteq \mathrm{span}(\mathbf{A})$; localizing in the reduced space ($\mathbf{A} = \mathbf{B}$) avoids the curse of dimensionality.

## 2.3. Empirical Optimization Problem

Let $(X_i, Y_i)_{i=1,\dots,n} \subset \mathbb{R}^p \times \mathbb{R}$ be i.i.d. random variables with common distribution $\mathbb{P}$. We estimate the local expectation $\mathbb{E}_{(x,u)}[\cdot]$ using the $k$-NN estimator: for a point $x$, we average over the set $\mathcal{N}_k(x)$ of the $k$ nearest neighbors to $x$ among $(X_1, \dots, X_n)$ with respect to the Euclidean norm.

Substituting into the integrated objective yields the empirical MAVE criterion: denote the local linear prediction residual by

$$r_{ij}(a, \mathbf{b}, \mathbf{B}) := Y_i - a - \mathbf{b}^\top \mathbf{B}^\top (X_i - X_j).$$

The empirical MAVE criterion is then

$$\min_{\mathbf{B} \in \mathrm{St}(p,d)} \left\{ D_n(\mathbf{B}) = \sum_{j=1}^n \sum_{i=1}^n r_{ij}(a_j, \mathbf{b}_j, \mathbf{B})^2\, w_{ij} \right\}. \tag{5}$$

where $w_{ij} = k^{-1}\mathbf{1}_{i \in \mathcal{N}_k(X_j)}$ are weights selecting neighbors of $X_j$, and for each $j$, $(a_j, \mathbf{b}_j)$ are local linear regression coefficients:

$$(a_j, \mathbf{b}_j) = \arg\min_{a, \mathbf{b}} \sum_{i=1}^{n} r_{ij}(a, \mathbf{b}, \mathbf{B})^2 \, w_{ij},$$

Different weight schemes may be used (e.g., Gaussian kernel weights). Moreover, following the cylinder relaxation of Remark 2.6, we may use $\mathcal{N}_{k,\mathbf{A}}(X_j)$ (the $k$-NN under the semi-norm $\|x\|_{\mathbf{A}} = \|\mathbf{A}^\top x\|_2$) for any $\mathbf{A}$ with $\mathrm{span}(\mathbf{B}) \subseteq \mathrm{span}(\mathbf{A})$. This leads to the adaptive procedure in Section 4.

We now reformulate (5) as a maximization problem amenable to Riemannian optimization. Define the local moment vector and Gram matrix

$$\mu^{(j)} = \sum_{i=1}^{n} w_{ij} \tilde{X}_i^{(j)} \tilde{Y}_i^{(j)} \in \mathbb{R}^p,$$

$$G^{(j)} = \sum_{i=1}^{n} w_{ij} \tilde{X}_i^{(j)} (\tilde{X}_i^{(j)})^\top \in \mathbb{R}^{p \times p},$$

where $\tilde{Y}_i^{(j)} = Y_i - \bar{Y}^{(j)}$, $\tilde{X}_i^{(j)} = X_i - \bar{X}^{(j)}$, and $\bar{Y}^{(j)}, \bar{X}^{(j)}$ are the weighted local means.

**Proposition 2.7** (Stiefel reformulation). *Assume $\mathbf{B}^\top G^{(j)} \mathbf{B}$ is invertible for every $j \in \{1, \dots, n\}$ and every $\mathbf{B} \in \mathrm{St}(p, d)$. Then minimizing (5) over $\mathbf{B} \in \mathrm{St}(p, d)$ is equivalent to maximizing*

$$F(\mathbf{B}) := \sum_{j=1}^{n} (\mu^{(j)})^\top \mathbf{B} (\mathbf{B}^\top G^{(j)} \mathbf{B})^{-1} \mathbf{B}^\top \mu^{(j)}. \quad (6)$$

*Moreover, $F(\mathbf{BQ}) = F(\mathbf{B})$ for any orthogonal $\mathbf{Q} \in \mathcal{O}(d)$, so $F$ descends to a well-defined function on the Grassmannian $\mathrm{Gr}(p, d)$.*

## 3. Related Work

We review gradient-based SDR methods and Riemannian optimization, situating our contributions within this literature.

**MAVE.** The original MAVE algorithm (Xia et al., 2002) minimizes the same objective $D_n$ but with kernel weights $w_{ij} \propto K_h(\|X_i - X_j\|)$ rather than $k$-NN, and employs a different optimization strategy. Rather than updating $\mathbf{B}$ globally, MAVE uses coordinate descent, cycling through columns $\boldsymbol{\beta}_1, \dots, \boldsymbol{\beta}_d$. Each column update requires solving a constrained quadratic subproblem subject to orthogonality with the remaining columns, itself handled by alternating between local coefficient updates and a KKT system. In contrast, our Riemannian approach updates all columns simultaneously, maintaining orthogonality intrinsically via retraction rather than explicit constraints. Despite well-understood statistical properties, the optimization of MAVE remains theoretically underdeveloped: coordinate descent lacks convergence guarantees except when a $\sqrt{n}$-consistent initialization is available (Xia, 2007), and no existing procedure provides convergence results for the underlying non-convex optimization problem.

Extensions include dMAVE (Xia, 2007), which transforms $Y$ through a density kernel to estimate the full central subspace rather than just the central mean subspace. Our framework could accommodate this extension, but we leave it for future work.

**OPG.** The Outer Product of Gradients method (Xia et al., 2002) estimates gradients via local linear regression in the ambient space. For each anchor $X_j$, OPG solves

$$\min_{a_j, b_j \in \mathbb{R}^p} \sum_{i=1}^{n} \big(Y_i - a_j - b_j^\top (X_i - X_j)\big)^2 w_{ij},$$

with kernel weights $w_{ij} \propto K_h(\|X_i - X_j\|)$ computed in $\mathbb{R}^p$. The gradient estimate is $\hat{b}_j = (G^{(j)})^{-1} \mu^{(j)}$, and the EDR subspace is recovered as the top $d$ eigenvectors of

$$\hat{\Sigma}_{\mathrm{OPG}} = \frac{1}{n} \sum_{j=1}^{n} \hat{b}_j \hat{b}_j^\top.$$

OPG admits a closed-form solution but computes neighborhoods in $\mathbb{R}^p$, suffering from the curse of dimensionality when $p$ is large. When local covariances are isotropic ($G^{(j)} \propto I_p$), OPG and MAVE coincide (Proposition C.1). A $k$-NN variant of OPG is proposed in (Ausset et al., 2021), which reduces computational cost by restricting computations to selected neighbors.

**RMAVE.** Refined MAVE (Xia et al., 2002), as well as the structure-adaptive approach of Hristache et al. (2001); Dalalyan & Spokoiny (2008), addresses the dimensionality issue through iterative refinement. Given an estimate $\widehat{\mathbf{B}}$, RMAVE recomputes kernel weights using proximity in the reduced space:

$$w_{ij} \propto K_h\big(\|\widehat{\mathbf{B}}^\top (X_i - X_j)\|\big),$$

with bandwidth chosen according to the reduced dimension $d$ rather than the ambient dimension $p$, so that smaller bandwidths become feasible.

However, RMAVE inherits MAVE's coordinate-descent structure and adds significant overhead: each outer iteration recomputes the full $n \times n$ matrix of pairwise kernel weights and runs a column-wise coordinate-descent solver over the $d$ columns of $\mathbf{B}$. Both steps are at least

quadratic in $n$. Crucially, RMAVE relies on OPG for initialization; when $p$ is large, this initialization is poor, limiting RMAVE's effectiveness even with reduced-space refinement.

This quadratic-in-$n$ bottleneck is shared by most existing gradient-based SDR methods, including the structure-adaptive refinements above. Our approach combines RMAVE's reduced-space localization with sparse $k$-NN weights and Riemannian optimization on $\mathrm{St}(p,d)$.

**MADE.** Minimum Average Deviance Estimation (Adragni, 2018) extends MAVE to exponential family distributions by replacing the squared loss with local deviance, enabling application to binary and count responses. For continuous responses with Gaussian errors, MADE with identity link reduces to the same optimization problem as the one of MAVE and SMAVE. Beyond this statistical equivalence, MADE and SMAVE address orthogonal challenges: MADE generalizes the loss function while SMAVE addresses computational scalability. The MADE algorithm relies on Riemannian conjugate gradient with matrix exponentials for geodesic retraction, parallel transport of tangent vectors, and Newton–Raphson iterations for local parameters, primitives that are orders of magnitude more expensive than our first-order stochastic updates with QR retraction.

**Kernel-based methods.** Kernel Dimension Reduction (Fukumizu et al., 2009) characterizes the central subspace via conditional covariance operators in reproducing kernel Hilbert spaces, avoiding explicit gradient estimation. While theoretically elegant, these methods involve different computational trade-offs (kernel matrix operations vs. local regression) and are not directly comparable to the MAVE family we focus on here.

**Riemannian optimization.** Riemannian optimization (Absil et al., 2008; Boumal, 2023) extends gradient-based methods to smooth manifolds, requiring two ingredients: the Riemannian gradient and a retraction mapping tangent vectors back to the manifold. Convergence guarantees for Riemannian SGD are well established (Bonnabel, 2013; Zhang et al., 2016). Appendix A provides background on Riemannian geometry.

## 4. SMAVE Algorithm

We now present SMAVE, which resolves the tension between statistical efficiency and computational scalability by: (i) replacing dense kernel weights with sparse $k$-NN weights in the projected space, and (ii) optimizing directly on $\mathrm{St}(p,d)$ via Riemannian stochastic gradient ascent.

### 4.1. Adaptive Sparse Localization via $k$-NN

RMAVE computes weights $w_{ij} \propto K_h(\|\widehat{\mathbf{B}}^\top(X_i - X_j)\|)$ for all $n^2$ pairs, though local regression requires only local information. We exploit this sparsity by defining

$$w_{ij} = \begin{cases} 1/k & \text{if } X_i \in \mathcal{N}_k(\widehat{\mathbf{B}}, X_j), \\ 0 & \text{otherwise,} \end{cases} \tag{7}$$

where $\mathcal{N}_k(\mathbf{B}, X_j)$ denotes the $k$ nearest neighbors of $X_j$ under the projected distance $\|\widehat{\mathbf{B}}^\top(\cdot - X_j)\|$. Since neighborhoods depend on $\widehat{\mathbf{B}}$, they become stale as the subspace estimate evolves. We rebuild the $k$-NN index every $\tau$ iterations via KD-tree on $\{\widehat{\mathbf{B}}_t^\top X_i\}_{i=1}^n$, tree construction costs $O(dn \log n)$. Between rebuilds, the fixed index introduces slight mismatch but avoids per-iteration overhead. Moderate values of $\tau$ balance gradient fidelity against computational cost (see Section 5).

### 4.2. Riemannian Stochastic Gradient Ascent

The following result, proved in Appendix D, provides the gradient required for optimization on $\mathrm{St}(p,d)$.

**Proposition 4.1** (Riemannian gradient). *Assume $\mathbf{B}^\top G^{(j)}\mathbf{B}$ is invertible for all $j$. The Riemannian gradient of $F$ in* (6) *is*

$$\mathrm{grad}\, F(\mathbf{B}) = 2\sum_{j=1}^n \left(\mu^{(j)} - G^{(j)}\mathbf{B}u^{(j)}\right)(u^{(j)})^\top,$$

*where* $u^{(j)} = (\mathbf{B}^\top G^{(j)}\mathbf{B})^{-1}\mathbf{B}^\top \mu^{(j)}$.

At each iteration $t$, we sample uniformly without replacement a mini-batch $J_t \subset \{1, \ldots, n\}$, independently of the past, and form the stochastic gradient

$$\mathbf{g}_t = \frac{2n}{|J_t|}\sum_{j \in J_t}\left(\mu^{(j)} - G^{(j)}\mathbf{B}_t u^{(j)}\right)(u^{(j)})^\top, \tag{8}$$

which is an unbiased estimator of the full gradient: $\mathbb{E}[\mathbf{g}_t \mid \mathbf{B}_t] = \mathrm{grad}\, F(\mathbf{B}_t)$. We normalize $\mathbf{g}_t \leftarrow \mathbf{g}_t/\|\mathbf{g}_t\|_F$ to stabilize step sizes, incorporate heavy-ball momentum, and update via QR retraction:

$$\mathbf{v}_{t+1} = \beta\mathbf{v}_t + \mathbf{g}_t, \qquad \mathbf{B}_{t+1} = R_{\mathbf{B}_t}^{\mathrm{QR}}(\alpha_t \mathbf{v}_{t+1}), \tag{9}$$

where $\alpha_t = \alpha_0/(1+\gamma t)$ is a decaying step size and $\beta$ a momentum coefficient. The neighborhood size is calibrated to match Silverman's effective kernel neighborhood in $\mathbb{R}^d$ (Section 5).

SMAVE procedure is summarized in Algorithm 1. Observe that SMAVE initializes from a uniform draw on $\mathrm{St}(p,d)$, obtained as the orthonormal factor of a $p \times d$ Gaussian matrix; this contrasts with RMAVE, which warm-starts from

**Algorithm 1** SMAVE

---

**Require:** Data $\{(X_i, Y_i)\}_{i=1}^n$, target dimension $d \leq p$, iterations $T$, refresh period $\tau$, mini-batch size $m \geq 1$
1: Initialize $\mathbf{B}_0 \in \mathrm{St}(p, d)$; set $k \leftarrow \lceil n^{4/(d+4)} \rceil$ clipped to $[20, n/3]$
2: Build $k$-NN index on $\{\mathbf{B}_0^\top X_i\}_{i=1}^n$
3: **for** $t = 0, \ldots, T-1$ **do**
4:     Draw $J_t \subset \{1, \ldots, n\}$, $|J_t| = m$, uniformly without replacement and independently of $(\mathbf{B}_0, J_0, \ldots, J_{t-1})$
5:     Compute $\mathbf{g}_t$ via (8); normalize $\mathbf{g}_t \leftarrow \mathbf{g}_t / \|\mathbf{g}_t\|_F$
6:     Update $\mathbf{B}_{t+1}$ via (9)
7:     **if** $t > 0$ **and** $t \bmod \tau = 0$ **then**
8:         Rebuild $k$-NN index on $\{\mathbf{B}_{t+1}^\top X_i\}_{i=1}^n$
9:     **end if**
10: **end for**
11: **return** $\mathbf{B}_T$

---

the OPG estimate. The combination of random initialization and stochastic updates decouples SMAVE from the OPG basin, a property that becomes critical when $p$ is large and OPG's ambient-space localization degrades.

*Remark* 4.2 (Momentum implementation). The momentum vector $\mathbf{v}_t$ accumulates in ambient space without parallel or vector transport between tangent spaces. This common simplification avoids $O(pd^2)$ transport costs per iteration; the QR retraction implicitly projects updates onto the manifold (Bécigneul & Ganea, 2018).

### 4.3. Convergence Analysis

We analyze **SMAVE** in a simplified regime that isolates the geometric and statistical contributions but does not match the implementation: fixed neighborhoods ($\tau = \infty$), no momentum ($\beta = 0$), no gradient normalization. Mini-batches are drawn uniformly without replacement, as in the implementation. We work with the regularized objective

$$F_\varepsilon(\mathbf{B}) = \sum_{j=1}^n (\mu^{(j)})^\top \mathbf{B} (\mathbf{B}^\top G_\varepsilon^{(j)} \mathbf{B})^{-1} \mathbf{B}^\top \mu^{(j)}, \quad (10)$$

where $G_\varepsilon^{(j)} := G^{(j)} + \varepsilon I_p$ for $\varepsilon > 0$, to handle the rank-deficient regime $p > k - 1$. This corresponds to ridge-penalized local regression (Remark D.4). The relation $\mathbf{B}^\top G_\varepsilon^{(j)} \mathbf{B} \succeq \varepsilon I_d$ on $\mathrm{St}(p, d)$ makes the inversion well-posed, and the closed form of Proposition 4.1 carries over to $\mathrm{grad}\, F_\varepsilon$. We first establish almost-sure convergence, then quantify the non-asymptotic rate.

**Proposition 4.3** (Asymptotic convergence)**.** *For step sizes $\alpha_t = \alpha_0/(1 + \gamma t)$ with $\alpha_0, \gamma > 0$ and $\alpha_0$ small enough (explicit threshold in Appendix D), almost surely: (i) $F_\varepsilon(\mathbf{B}_t)$ converges to a finite limit; (ii) $\mathrm{grad}\, F_\varepsilon(\mathbf{B}_t) \to 0$; (iii) every accumulation point of $\{\mathbf{B}_t\}$ is a critical point of $F_\varepsilon$.*

Asymptotic convergence leaves the rate open. Under a constant step size, the following gives an explicit non-asymptotic bound on the average squared gradient.

**Theorem 4.4** (Non-asymptotic rate)**.** *There exist explicit constants $\bar{\alpha}_\varepsilon, L_\varepsilon, \sigma_\varepsilon^2, \Delta_\varepsilon$ (see Appendix D.4) such that, for any constant step size $\alpha \in (0, \bar{\alpha}_\varepsilon]$ and any $T \geq 1$,*

$$\frac{1}{T} \sum_{t=0}^{T-1} \mathbb{E}\big[\|\mathrm{grad}\, F_\varepsilon(\mathbf{B}_t)\|_F^2\big] \leq \frac{2\Delta_\varepsilon}{\alpha T} + L_\varepsilon \alpha \sigma_\varepsilon^2. \quad (11)$$

*Tuning $\alpha \propto 1/\sqrt{T}$ yields the rate $O(1/\sqrt{T})$.*

The results and their proofs (with explicit constants) appear in Appendix D. Proposition 4.3 follows from Bonnabel (2013, Thm.2). Theorem 4.4 combines retraction smoothness of $F_\varepsilon$ (the sole geometric input, yielding the per-step descent inequality in Boumal et al., 2019) with the without-replacement finite-population variance. The $T$-dependence matches the rate of Ghadimi & Lan (2013) for Euclidean stochastic gradient methods, which is optimal among stochastic first-order methods on non-convex smooth objectives under bounded variance (Arjevani et al., 2023).

The analysis applies verbatim to any localization scheme (kernel, $k$-NN, fixed-radius, similarity graphs) yielding an objective of the form (10): the Riemannian-optimization and statistical layers decouple. The $\varepsilon$-degradation of the constants and its consequences are deferred to the Appendix discussion. The convergence analysis of the full SMAVE algorithm, with periodic $k$-NN refresh inducing time-varying objectives and heavy-ball momentum inducing temporal correlations, remains open.

## 5. Numerical Experiments

### 5.1. Baselines

**Methods.** We focus the main comparison on **OPG**, **RMAVE** (Section 3), and our **SMAVE**. Two further methods, **OPG-$k$NN** and **RMAVE-$k$NN**, replace the kernel weights of OPG and RMAVE with the same uniform $k$-NN weights as SMAVE (in $\mathbb{R}^p$ and $\mathbb{R}^d$ respectively); these are diagnostic ablations isolating the effect of sparse localization, and we defer their results to Appendix F. We omit structure-adaptive OPG (Hristache et al., 2001; Dalalyan et al., 2008), which improves statistical efficiency but shares RMAVE's quadratic-in-$n$ refinement cost.

**Neighborhood calibration.** Let $q$ denote the localization dimension: $q = p$ for ambient-space methods (OPG, OPG-$k$NN) and $q = d$ for projected-space methods (RMAVE, RMAVE-$k$NN, SMAVE). Kernel methods use Gaussian weights with Silverman's rule-of-thumb bandwidth $h \propto n^{-1/(q+4)}$ (Silverman, 2018); $k$-NN methods use $k =$

$\lceil n^{4/(q+4)} \rceil$, matching the expected number of points within Silverman's effective kernel support. Full implementation details are in Appendix E.

**Outer iteration vs gradient steps.** RMAVE and SMAVE differ in what an iteration computes. One RMAVE *outer* iteration recomputes the full $n \times n$ matrix of pairwise kernel weights from the current subspace estimate and runs a column-wise coordinate-descent (CD) solver cycling over the $d$ columns of $\mathbf{B}$, each column update itself alternating between local-coefficient and direction updates. One SMAVE iteration is a single Riemannian gradient step on $\mathbf{B}$ using a mini-batch of $m$ anchors with $k$-NN weights.

**Per-iteration complexity.** For OPG, the dominant costs are forming the $n$ local Gram matrices $G^{(j)} \in \mathbb{R}^{p \times p}$ ($O(n^2 p^2)$) and solving the $n$ local $p$-dimensional systems ($O(np^3)$). OPG-$k$NN restricts each local regression to its $k$ neighbors, giving $O(nkp^2 + np^3)$, excluding KD-tree, construction and querying, which for reasonable dimension $p$ is secondary. For RMAVE, one outer iteration computes the $n^2$ projected pairwise distances in $\mathbb{R}^d$ ($O(n^2 d)$) and reforms all $n$ local Gram matrices from the resulting dense weights ($O(n^2 p^2)$); the latter dominates in the regime $p \gg d$ relevant to SDR, but the subsequent CD updates are subdominant in our implementation. RMAVE-$k$NN replaces the dense weights with $k$-NN weights at cost $O(nkp^2)$ but leaves the CD solver without exploiting weight sparsity. The CD solver cost remains unchanged and dominates overall. This is consistent with our finding (Appendix F). For SMAVE, the per-iteration cost is dominated by the computation of $m$ local Gram matrices of size $p \times p$ leading to an $O(mkp^2)$ overhead per iteration, to which KD-tree construction and querying ($O(n \log(n))$, when $d$ is small), should be added. The advantage of SMAVE is thus to bypass the CD solver of RMAVE, replacing it with a cheap stochastic gradient step that exploits small minibatch sizes $m$ and sparse $k$-NN weights.

### 5.2. Synthetic Experiments

We consider the SDR model $Y = g(\mathbf{B}_*^\top X) + \xi$, where $X \sim \mathcal{N}(0, \mathbf{\Sigma})$, $\mathbf{B}_* \in \text{St}(p, d)$ is a sparse EDR matrix (20% nonzero entries, QR-orthonormalized) with $d = 2$, and $\xi \sim \mathcal{N}(0, 0.5^2)$.

Sparse EDR matrices arise naturally in high-dimensional settings where only a subset of covariates contributes to the response, aligning with the sufficient variable selection literature (Li, 2007). All methods compared are agnostic to the sparsity structure and do not exploit it explicitly.

**Evaluation metric.** Given the true EDR matrix $\mathbf{B}_*$ and an estimate $\widehat{\mathbf{B}}$, we measure accuracy via the squared sub-

*Table 1.* Experimental design: 10 scenarios from 5 link functions $\times$ 2 covariance structures, with $n \in \{1000, 2000, 5000\}$ and $p \in \{50, 100, 200\}$.

| Factor | Levels |
|---|---|
| Link functions | Polynomial, Sinusoidal, Exponential, Interaction, Rational |
| Covariance $\mathbf{\Sigma}$ | Identity ($\mathbf{I}_p$); AR(1) ($\Sigma_{ij} = 0.5^{|i-j|}$) |
| Sample size $n$ | 1000, 2000, 5000 |
| Dimension $p$ | 50, 100, 200 |
| Replications | 10 per configuration |

space distance

$$m^2(\widehat{\mathbf{B}}, \mathbf{B}_*) = \left\| (I_p - \mathbf{B}_* \mathbf{B}_*^\top) \widehat{\mathbf{B}} \right\|_F^2, \qquad (12)$$

which equals zero when the column spaces coincide and is bounded by $d$ when they are orthogonal.

*Table 2.* Squared subspace distance $m^2$ (mean $\pm$ s.e. over 10 scenarios $\times$ 10 replications). **Bold**: best; underline: second best.

| $n$ | $p$ | OPG | RMAVE | SMAVE |
|---|---|---|---|---|
| | 50 | $1.90_{\pm.01}$ | $0.54_{\pm.05}$ | $\mathbf{0.25_{\pm.03}}$ |
| 1000 | 100 | $1.95_{\pm.00}$ | $\underline{0.98_{\pm.04}}$ | $\mathbf{0.59_{\pm.04}}$ |
| | 200 | $1.96_{\pm.01}$ | $\underline{1.48_{\pm.03}}$ | $\mathbf{1.02_{\pm.04}}$ |
| | 50 | $1.89_{\pm.01}$ | $0.43_{\pm.04}$ | $\mathbf{0.11_{\pm.02}}$ |
| 2000 | 100 | $1.95_{\pm.00}$ | $\underline{0.73_{\pm.04}}$ | $\mathbf{0.28_{\pm.03}}$ |
| | 200 | $1.98_{\pm.00}$ | $\underline{1.12_{\pm.04}}$ | $\mathbf{0.59_{\pm.04}}$ |
| | 50 | $1.89_{\pm.01}$ | $0.33_{\pm.04}$ | $\mathbf{0.03_{\pm.01}}$ |
| 5000 | 100 | $1.94_{\pm.00}$ | $\underline{0.40_{\pm.05}}$ | $\mathbf{0.13_{\pm.03}}$ |
| | 200 | $1.98_{\pm.00}$ | $\underline{0.69_{\pm.05}}$ | $\mathbf{0.25_{\pm.03}}$ |

*Table 3.* Runtime in seconds (mean over 10 scenarios $\times$ 10 replications). **Bold**: fastest; underline: second fastest.

| $n$ | $p$ | OPG | RMAVE | SMAVE |
|---|---|---|---|---|
| | 50 | $\underline{0.29}$ | 3.86 | $\mathbf{0.28}$ |
| 1000 | 100 | $\underline{0.68}$ | 5.83 | $\mathbf{0.49}$ |
| | 200 | $\underline{1.75}$ | 9.99 | $\mathbf{1.41}$ |
| | 50 | $\underline{1.04}$ | 12.94 | $\mathbf{0.35}$ |
| 2000 | 100 | $\underline{2.41}$ | 20.89 | $\mathbf{0.69}$ |
| | 200 | $\underline{6.25}$ | 34.89 | $\mathbf{1.78}$ |
| | 50 | $\underline{6.44}$ | 61.05 | $\mathbf{0.95}$ |
| 5000 | 100 | $\underline{14.16}$ | 98.01 | $\mathbf{2.02}$ |
| | 200 | $\underline{37.14}$ | 173.88 | $\mathbf{5.12}$ |

**Subspace recovery accuracy.** Tables 2–3 report the squared subspace distance $m^2$ and runtime, aggregated across all 10 scenarios. The ranking depends on $p$, and the transition is informative.

At $p \in \{50, 100, 200\}$, SMAVE reduces error by 2–9$\times$ compared to RMAVE while running an order of magnitude

faster. The bottleneck is initialization quality: RMAVE warm-starts from the OPG solution, whose kernel bandwidth scales as $h \sim n^{-1/(p+4)}$ in $\mathbb{R}^p$ and degrades exponentially as $p$ grows. RMAVE's deterministic coordinate descent then refines this starting point but cannot escape suboptimal basins. SMAVE's three coupled design choices address this regime directly: (i) $k$-NN neighborhoods in the *projected* space $\mathbb{R}^d$ rather than the ambient space $\mathbb{R}^p$, eliminating the curse of dimensionality in the localization step; (ii) uniform initialization on $\mathrm{St}(p, d)$, removing the dependence on the OPG warm start; and (iii) mini-batch stochastic Riemannian gradients, which explore the non-convex landscape rather than committing early to one basin. Figure 4 (Appendix F.6) makes the landscape effect visible: RMAVE starts with lower error but plateaus early, while SMAVE continues improving from random initialization throughout the 100 iterations; the wider confidence bands for SMAVE reflect stochastic gradient variance, but this is precisely what enables escape from suboptimal basins.

In the lower-dimensional regime (Appendix F, $p \in \{10, 20\}$), RMAVE matches or slightly beats SMAVE. Here the OPG initialization is already accurate ($h \sim n^{-1/(p+4)}$ is effective when $p$ is small) so the optimization-landscape advantages of SMAVE no longer dominate, and a secondary effect surfaces: RMAVE's smooth Gaussian weights produce better-calibrated local regressions than SMAVE's hard $k$-NN indicators. This is a minor disadvantage at low $p$ that becomes inconsequential as $p$ grows and initialization quality takes over.

**Accuracy–runtime tradeoff.** Table 3 reveals the cost of statistical accuracy. RMAVE achieves reasonable estimates but its per-outer-iteration cost scales as $O(n^2 p^2)$ in our implementation, dominated by the $n$ local $p \times p$ Gram matrices; multiplied across $T$ outer iterations this becomes prohibitive at scale. At $(n, p) = (5000, 200)$, RMAVE takes nearly three minutes per replication while SMAVE finishes in roughly five seconds. OPG offers a closed-form solution at lower cost but substantially worse accuracy. SMAVE resolves this tradeoff, matching or exceeding RMAVE's accuracy at all reported $(n, p)$ configurations while running 10–35× faster. Figure 1 illustrates the resulting Pareto frontier.

### 5.3. Real-data Experiments

We evaluate SDR methods on real regression tasks where the true subspace is unknown and performance is measured by downstream prediction accuracy.

**Datasets.** Table 4 summarizes four publicly available datasets spanning diverse domains and scales. **Wine Quality** (Cortez et al., 2009) combines red and white wine samples, predicting sensory quality scores from 11 physic-

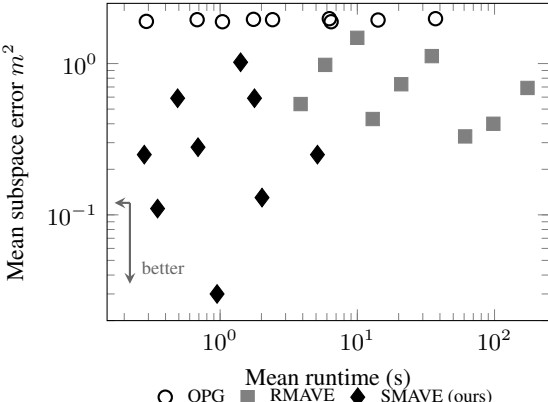

*Figure 1.* Accuracy–efficiency trade-off ($p \in \{50, 100, 200\}$, $n \in \{1000, 2000, 5000\}$). SMAVE consistently occupies the favorable lower-left region.

*Table 4.* Real-world datasets for prediction experiments.

| Dataset | $n$ | $p$ | $d$ candidates |
|---|---|---|---|
| Wine Quality | 6,497 | 11 | $\{2, 4, 6, 8\}$ |
| Seoul Bike | 8,760 | 14 | $\{2, 4, 6, 8, 10\}$ |
| Pumadyn | 8,192 | 32 | $\{2, 4, 8, 12, 16\}$ |
| Gas Sensor | 2,565 | 128 | $\{5, 10, 20, 40, 60\}$ |

ochemical properties. **Seoul Bike** (Sathishkumar et al., 2020) predicts hourly rental demand from weather conditions and temporal features. **Pumadyn** (Orr et al., 2000) predicts angular acceleration from 32 attributes of simulated robot arm dynamics. **Gas Sensor** (Fonollosa et al., 2015) predicts gas concentration from a 128-dimensional metal-oxide sensor array, providing a high-dimensional test case where the curse of dimensionality is most pronounced.

**Evaluation protocol.** We perform 5-fold cross-validation repeated 3 times, yielding 15 train/test splits per configuration. Features and response are centered and scaled using statistics computed exclusively on the training fold. The test fold is then transformed using these training statistics. For each candidate reduced dimension $d$, we apply the SDR method to training data, project both splits via $Z = X\widehat{\mathbf{B}}$, and fit a distance-weighted $k$-NN regressor with $k = \min(10, \lfloor n_{\text{train}}/5 \rfloor)$ on the projected training data. We select $d^* = \arg\min_d \overline{\mathrm{MSE}}_{\mathrm{CV}}(d)$ based on mean test MSE across all 15 splits and report the corresponding performance.

The candidate dimensions $d$ in Table 4 are chosen to span a range from aggressive reduction to near-full rank, scaled to each dataset's ambient dimension $p$. For low-dimensional datasets ($p \leq 15$), candidates cover most of the range up to $p - 1$. For moderate dimensions ($p \in [26, 32]$), candidates extend to roughly $p/2$. For the high-dimensional Gas Sensor dataset ($p = 128$), candidates range from 5 to 60, test-

*Table 5.* Test MSE (mean $\pm$ s.d. over 15 splits) and selected dimension $d^*$. **Bold**: best MSE; underline: second best.

| Dataset | Method | MSE | $d^*$ | Time (s) |
|---|---|---|---|---|
| Wine Quality | PCA | $0.515 \pm 0.015$ | 8 | $< 0.1$ |
| $n=6497$ | OPG | $0.521 \pm 0.018$ | 8 | 1.4 |
| $p=11$ | RMAVE | $\mathbf{0.498 \pm 0.017}$ | 8 | 600.2 |
| | SMAVE | $0.511 \pm 0.018$ | 8 | 0.2 |
| Seoul Bike | PCA | $0.215 \pm 0.015$ | 10 | $< 0.1$ |
| $n=8760$ | OPG | $0.268 \pm 0.012$ | 8 | 2.8 |
| $p=14$ | RMAVE | $\mathbf{0.165 \pm 0.008}$ | 4 | 327.3 |
| | SMAVE | $0.167 \pm 0.011$ | 4 | 0.3 |
| Pumadyn | PCA | $1.070 \pm 0.048$ | 16 | $< 0.1$ |
| $n=8192$ | OPG | $0.174 \pm 0.009$ | 2 | 4.4 |
| $p=32$ | RMAVE | $\mathbf{0.092 \pm 0.003}$ | 2 | 55.4 |
| | SMAVE | $0.127 \pm 0.014$ | 4 | 0.5 |
| Gas Sensor | PCA | $0.020 \pm 0.011$ | 10 | $< 0.1$ |
| $n=2565$ | OPG | $0.026 \pm 0.014$ | 20 | 1.9 |
| $p=128$ | RMAVE | $0.025 \pm 0.013$ | 5 | 35.1 |
| | SMAVE | $\mathbf{0.018 \pm 0.011}$ | 5 | 0.2 |

*Table 6.* Direct comparison of SMAVE and RMAVE for sufficient dimension reduction. Test MSE (mean $\pm$ std) computed over 15 independent cross-validation splits. The $p$-value column reports paired $t$-test results; bold marks the method with significantly lower error ($p < 0.05$).

| Dataset | RMAVE | SMAVE | $p$-value |
|---|---|---|---|
| Wine Quality | $\mathbf{0.498 \pm 0.017}$ | $0.511 \pm 0.018$ | $9.2 \times 10^{-5}$ |
| Seoul Bike | $0.165 \pm 0.008$ | $0.167 \pm 0.011$ | 0.47 |
| Pumadyn | $\mathbf{0.092 \pm 0.003}$ | $0.127 \pm 0.014$ | $9.4 \times 10^{-8}$ |
| Gas Sensor | $0.025 \pm 0.013$ | $\mathbf{0.018 \pm 0.011}$ | 0.027 |

ing whether SDR methods can identify a low-dimensional structure amid many irrelevant features. This design ensures fair comparison: all methods search over the same candidate set, and the best $d^*$ is selected independently for each method.

We compare OPG, RMAVE, and SMAVE from Section 3, along with PCA as an unsupervised baseline. RMAVE and SMAVE run for $T = 100$ iterations. SMAVE hyperparameters are fixed to the values tuned on synthetic data (Appendix E), with no dataset-specific adjustments.

**Results.** Figure 5 (Appendix G) shows test MSE across all candidate dimensions; Table 5 reports results at the optimal $d^*$ selected independently per method by cross-validation. Test MSE is computed on standardized responses to enable cross-dataset comparison.

**Analysis.** Table 6 provides a direct comparison between SMAVE and RMAVE. RMAVE achieves significantly lower error on Wine Quality ($p = 9.2 \times 10^{-5}$) and Pumadyn ($p = 9.4 \times 10^{-8}$), though the absolute differences are modest (0.013 and 0.035, respectively) while runtime dif-

fers by 2–3 orders of magnitude. On Seoul Bike, the two methods are statistically indistinguishable ($p = 0.47$). On Gas Sensor ($p = 128$), SMAVE significantly outperforms RMAVE ($p = 0.027$), consistent with the synthetic finding that the stochastic approach scales better in high ambient dimension.

Both MAVE-based methods consistently select more parsimonious subspaces than OPG: $d^* = 5$ on Gas Sensor versus $d^* = 20$ for OPG, achieving lower error with $4\times$ fewer dimensions. This pattern suggests the MAVE objective better identifies the minimal sufficient subspace.

## 6. Conclusion

We introduced SMAVE, a scalable algorithm combining Riemannian stochastic optimization on the Stiefel manifold with adaptive projected-space $k$-NN localization, and showed that MAVE shares its target with OPG via projected-space local regression. SMAVE matches or improves on RMAVE's subspace recovery at moderate-to-high ambient dimension while running much faster, with the gap widening as $p$ grows. The theory developed in this work opens several natural directions for future research. First, an explicit deviation inequality for the empirical risk minimizer could be derived, extending Proposition 2.5 to the estimation setting. Second, extending the analysis to iterate-dependent neighborhoods, as arising in SMAVE, would provide finite-sample guarantees for this practically important algorithm.

The scalability techniques developed here could potentially be combined with other objective functions in future work. Two concrete directions come to mind. First, the MADE deviance-based objective function (Adragni, 2018) would be of interest for non-Gaussian regression model. Second, SMAVE does not readily extend to the classification setting, as its objective function is tailored to regression. The local logistic risk objective recently proposed in (Ahmad et al., 2024) allows gradient estimation of the conditional probability function. Both represent promising avenues for extending our stochastic gradient descent approach.

## Acknowledgements

This research work is supported by France 2030 funding managed by the National Research Agency (ANR) as part of IA CLUSTER program, reference ANR-23-IACL-0003 - DATAIA CLUSTER.

## Impact Statement

SMAVE is a general-purpose statistical method without direct harm capabilities. However, the learned projection may retain sensitive attributes correlated with the response, or discard fairness-relevant features orthogonal to it. On the positive side, linear projections are inherently auditable, and SMAVE's computational gains lower the barrier to principled dimensionality reduction in resource-constrained settings.

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

# A. Riemannian Geometry Background

We provide the necessary background on Riemannian optimization over the Stiefel manifold. For a comprehensive treatment, we refer the reader to Absil et al. (2008); Boumal (2023).

**The Stiefel and Grassmann manifolds.** The Stiefel manifold $\mathrm{St}(p, d)$ is the set of $p \times d$ matrices with orthonormal columns:

$$\mathrm{St}(p, d) = \left\{ \mathbf{B} \in \mathbb{R}^{p \times d} : \mathbf{B}^\top \mathbf{B} = I_d \right\}.$$

It is a compact embedded submanifold of $\mathbb{R}^{p \times d}$ of dimension $pd - d(d+1)/2$.

The Grassmann manifold $\mathrm{Gr}(p, d)$ is the set of all $d$-dimensional linear subspaces of $\mathbb{R}^p$. Since two matrices $\mathbf{B}, \mathbf{B}' \in \mathrm{St}(p, d)$ span the same subspace if and only if $\mathbf{B}' = \mathbf{B}\mathbf{Q}$ for some orthogonal matrix $\mathbf{Q} \in \mathcal{O}(d)$, the Grassmann manifold can be identified with the quotient

$$\mathrm{Gr}(p, d) \cong \mathrm{St}(p, d)/\mathcal{O}(d).$$

We therefore optimize over $\mathrm{St}(p, d)$ as a convenient computational representation, with the understanding that any two solutions differing by a right orthogonal factor represent the same subspace.

**Tangent space and Riemannian metric.** The tangent space to $\mathrm{St}(p, d)$ at $\mathbf{B}$ is

$$T_\mathbf{B}\mathrm{St}(p, d) = \left\{ \xi \in \mathbb{R}^{p \times d} : \mathbf{B}^\top \xi + \xi^\top \mathbf{B} = 0 \right\},$$

which consists of matrices whose product with $\mathbf{B}^\top$ is skew-symmetric. Throughout the paper, $\mathrm{St}(p, d)$ is equipped with the *embedded Frobenius metric* inherited from $\mathbb{R}^{p \times d}$, that is, $\langle \xi, \eta \rangle_\mathbf{B} = \mathrm{tr}(\xi^\top \eta)$ for $\xi, \eta \in T_\mathbf{B}\mathrm{St}(p, d)$. All Riemannian gradients, projections, and norms below refer to this metric; the alternative canonical metric of Edelman et al. (1998) would yield different expressions.

The orthogonal projection of $Z \in \mathbb{R}^{p \times d}$ onto $T_\mathbf{B}\mathrm{St}(p, d)$ (Absil et al., 2008, §3.6.1) is

$$\Pi_{T_\mathbf{B}\mathrm{St}}(Z) = Z - \mathbf{B}\,\mathrm{sym}(\mathbf{B}^\top Z), \qquad \mathrm{sym}(A) = \tfrac{1}{2}(A + A^\top).$$

**Euclidean and Riemannian gradients.** We adopt the following convention throughout the paper. Let $f : \mathcal{U} \to \mathbb{R}$ be a smooth function defined on an open set $\mathcal{U} \subset \mathbb{R}^{p \times d}$ containing $\mathrm{St}(p, d)$. The *Euclidean* (or *ambient*) gradient of $f$ at $\mathbf{B}$ is the matrix of partial derivatives,

$$\nabla f(\mathbf{B}) \in \mathbb{R}^{p \times d}, \qquad \left( \nabla f(\mathbf{B}) \right)_{ij} = \frac{\partial f}{\partial \mathbf{B}_{ij}}(\mathbf{B}), \quad i \in \{1, \ldots, p\},\, j \in \{1, \ldots, d\}.$$

The *Riemannian* gradient of $f\big|_{\mathrm{St}(p,d)}$ at $\mathbf{B}$ is the unique tangent vector $\mathrm{grad}\, f(\mathbf{B}) \in T_\mathbf{B}\mathrm{St}(p, d)$ satisfying

$$df(\mathbf{B})[\xi] = \langle \mathrm{grad}\, f(\mathbf{B}), \xi \rangle_F \qquad \text{for all } \xi \in T_\mathbf{B}\mathrm{St}(p, d),$$

where $\langle \cdot, \cdot \rangle_F$ is the Frobenius inner product. Under the embedded Frobenius metric on $\mathrm{St}(p, d)$, the Riemannian gradient is the orthogonal projection of the Euclidean gradient onto the tangent space:

$$\mathrm{grad}\, f(\mathbf{B}) = \Pi_{T_\mathbf{B}\mathrm{St}}\big( \nabla f(\mathbf{B}) \big). \tag{13}$$

We emphasize that $\nabla f(\mathbf{B})$ requires $f$ to be defined on an ambient neighborhood of $\mathbf{B}$; functions defined only intrinsically on $\mathrm{St}(p, d)$ do not admit a Euclidean gradient without such an extension. In the present paper, every $f$ we differentiate (notably $F$ and $F_\varepsilon$) is naturally defined on an open set $\mathcal{U} \supset \mathrm{St}(p, d)$ in $\mathbb{R}^{p \times d}$, so no ambiguity arises.

The same symbol $\nabla$ is also used in Section 2 and Appendix B to denote the standard gradient of the regression function $g : \mathbb{R}^p \to \mathbb{R}$, that is, $\nabla g(x) \in \mathbb{R}^p$ with $(\nabla g(x))_i = \partial g/\partial x_i$. Context resolves which meaning applies: $\nabla g(x)$ is a vector in $\mathbb{R}^p$, while $\nabla f(\mathbf{B})$ is a matrix in $\mathbb{R}^{p \times d}$.

**Retractions.** The exponential map $\exp_x : T_x\mathcal{M} \to \mathcal{M}$ sends a tangent vector $\xi$ to the endpoint of the geodesic starting at $x$ with initial velocity $\xi$, providing the canonical way to move from a point along a tangent direction while staying on the manifold. A retraction $R : T\mathcal{M} \to \mathcal{M}$ relaxes this by requiring only first-order agreement with the exponential map: (i) $R_x(0_x) = x$, and (ii) $\frac{d}{dt}\big|_{t=0} R_x(t\xi) = \xi$ for all $\xi \in T_x\mathcal{M}$. These conditions suffice for convergence of gradient-based methods while being cheaper to compute. On the Stiefel manifold, we use the QR retraction, which approximates the exponential map at the first order. Given a matrix $A \in \mathbb{R}^{p \times d}$ with full column rank, let $A = QR$ be its thin QR decomposition, where $Q \in \mathrm{St}(p, d)$ and $R \in \mathbb{R}^{d \times d}$ is upper triangular with positive diagonal entries. We denote by $\mathrm{qf}(A) = Q$ the Q-factor of this decomposition. The QR retraction is then defined as

$$R_{\mathbf{B}}^{\mathrm{QR}}(\xi) = \mathrm{qf}(\mathbf{B} + \xi), \quad \xi \in T_{\mathbf{B}}\mathrm{St}(p, d),$$

which maps the tangent vector $\xi$ to a point on the manifold by orthonormalizing the columns of $\mathbf{B} + \xi$. This retraction has computational complexity $O(pd^2)$.

## B. Proofs of Section 2: SDR and Local Approximation

**Discussion on assumptions** Recall some details about the mathematical background of the paper. Let $(Y, X)$ be a random vector where $Y \in \mathbb{R}$ and $X \in \mathbb{R}^p$. We consider the regression function $g(x) := \mathbb{E}[Y \mid X = x]$ and write $Y = g(X) + \varepsilon$ where $\mathbb{E}[\varepsilon \mid X] = 0$ and $\mathrm{var}(\varepsilon \mid X) = \sigma^2(X)$.

For a matrix $B \in \mathbb{R}^{p \times d}$ with orthonormal columns (i.e., $B \in \mathrm{St}(p, d)$), we denote by $P_B = BB^\top$ the orthogonal projection onto the column space of $B$.

For $x \in \mathbb{R}^p$ and $u > 0$, let $\mathbb{E}_{(x,u)}[\cdot]$ denote the conditional expectation given that $X \in B(x, u) = \{y \in \mathbb{R}^p : \|y - x\| \leq u\}$. The local objective function is defined as:

$$D_{(x,u)}(B) := \min_{a \in \mathbb{R},\, b \in \mathbb{R}^d} \mathbb{E}_{(x,u)}\left[(Y - a - b^\top B^\top(X - x))^2\right]. \tag{14}$$

**Assumption B.1** ($L$-smoothness). There exists $L > 0$ such that, for $u$ small enough,

$$\mathbb{E}_{(x,u)}\left[|g(X) - g(x) - \nabla g(x)^\top(X - x)|^2\right] \leq \frac{L^2}{4}u^4. \tag{15}$$

**Assumption B.2** (Local covariance lower bound). There exists $\lambda \in (0, 1]$ such that, for $u$ small enough,

$$G_{(x,u)} \succeq \lambda u^2 I_p, \tag{16}$$

where $G_{(x,u)} := \mathbb{E}_{(x,u)}\left[(X - \mathbb{E}_{(x,u)}[X])(X - \mathbb{E}_{(x,u)}[X])^\top\right]$ is the local covariance matrix.

*On the constraint $\lambda \leq 1$:* For any real random variable $Y$ and any fixed $y \in \mathbb{R}$, the mean minimizes the expected squared deviation: $\mathbb{E}[(Y - \mathbb{E}[Y])^2] \leq \mathbb{E}[(Y - y)^2]$. Applying this to $Y = X^\top v$ for arbitrary $v \in \mathbb{R}^p$ and the fixed point $x^\top v$ yields $v^\top G_{(x,u)} v \leq v^\top \mathbb{E}_{(x,u)}[(X - x)(X - x)^\top]v$. Since this holds for all $v$, we obtain the matrix inequality $G_{(x,u)} \preceq \mathbb{E}_{(x,u)}[(X - x)(X - x)^\top]$. Under $\mathbb{E}_{(x,u)}$, we have $X \in B(x, u)$, so $\|X - x\| \leq u$ almost surely, which implies $\mathbb{E}_{(x,u)}[(X - x)(X - x)^\top] \preceq u^2 I_p$. Combining these inequalities: $G_{(x,u)} \preceq u^2 I_p$, hence $\lambda \leq 1$.

This section contains the proofs of the mathematical statements of the paper.

### B.1. Proof of Proposition 2.4

**Proof of upper bound.** We need to show that if $(I_p - P_B)\nabla g(x) = 0$ (i.e., $\nabla g(x) \in \mathrm{span}(B)$), then

$$D_{(x,u)}(B) \leq \mathbb{E}_{(x,u)}[\sigma^2(X)] + \frac{L^2}{4}u^4. \tag{17}$$

Since $Y = g(X) + \varepsilon$ with $\mathbb{E}[\varepsilon \mid X] = 0$, we have the decomposition:

$$\mathbb{E}_{(x,u)}\left[(Y - a - b^\top B^\top(X - x))^2\right] = \mathbb{E}_{(x,u)}[\sigma^2(X)] + \mathbb{E}_{(x,u)}\left[(g(X) - a - b^\top B^\top(X - x))^2\right]. \tag{18}$$

Indeed, expanding the square and using $\mathbb{E}_{(x,u)}[\varepsilon \cdot h(X)] = \mathbb{E}_{(x,u)}[h(X) \cdot \mathbb{E}[\varepsilon \mid X]] = 0$ for any function $h$.

Thus, minimizing over $(a, b)$ gives:

$$D_{(x,u)}(B) = \mathbb{E}_{(x,u)}[\sigma^2(X)] + \min_{a \in \mathbb{R},\, b \in \mathbb{R}^d} \mathbb{E}_{(x,u)}\left[(g(X) - a - b^\top B^\top (X - x))^2\right]. \tag{19}$$

Take $a = g(x)$ and $b = B^\top \nabla g(x)$. With this choice:

$$b^\top B^\top (X - x) = \nabla g(x)^\top B B^\top (X - x) = \nabla g(x)^\top P_B (X - x).$$

Now we use the gradient condition: If $(I_p - P_B)\nabla g(x) = 0$, then $P_B \nabla g(x) = \nabla g(x)$, so:

$$b^\top B^\top (X - x) = \nabla g(x)^\top (X - x).$$

By definition of the minimum:

$$\min_{a,b} \mathbb{E}_{(x,u)}\left[(g(X) - a - b^\top B^\top (X - x))^2\right] \leq \mathbb{E}_{(x,u)}\left[(g(X) - g(x) - \nabla g(x)^\top (X - x))^2\right] \leq \frac{L^2}{4} u^4,$$

where the last inequality uses Assumption B.1. □

**Proof of lower bound:** We need to show that whenever $uL \leq \frac{\lambda}{2}\|(I_p - P_B)\nabla g(x)\|_2$,

$$D_{(x,u)}(B) \geq \mathbb{E}_{(x,u)}[\sigma^2(X)] + \frac{\lambda u^2}{2}\|(I_p - P_B)\nabla g(x)\|_2^2. \tag{20}$$

We adopt the following strategy. We optimize with respect to $a$ first, characterize the optimal $b$, then bound from below.

*Step 1: Optimize over $a$.* Define the centered quantities:

$$\tilde{g}_{(x,u)} := g(X) - \mathbb{E}_{(x,u)}[g(X)], \tag{21}$$
$$\tilde{X}_{(x,u)} := X - \mathbb{E}_{(x,u)}[X]. \tag{22}$$

Optimizing over $a$ (taking $a^* = \mathbb{E}_{(x,u)}[g(X)] - b^\top B^\top \mathbb{E}_{(x,u)}[X - x]$):

$$\min_{a \in \mathbb{R}} \mathbb{E}_{(x,u)}\left[(g(X) - a - b^\top B^\top (X - x))^2\right] = \mathbb{E}_{(x,u)}\left[(\tilde{g}_{(x,u)} - b^\top B^\top \tilde{X}_{(x,u)})^2\right]. \tag{23}$$

*Step 2: Characterize the optimal $b$.* After optimizing over $a$, we need to solve:

$$\min_{b \in \mathbb{R}^d} L(b) := \mathbb{E}_{(x,u)}\left[(\tilde{g}_{(x,u)} - b^\top B^\top \tilde{X}_{(x,u)})^2\right], \tag{24}$$

where $B \in \mathrm{St}(p, d)$ is fixed.

*Step 2.1: Expand the objective.* Expanding the square:

$$L(b) = \mathbb{E}_{(x,u)}[\tilde{g}_{(x,u)}^2] - 2b^\top B^\top \mathbb{E}_{(x,u)}[\tilde{X}_{(x,u)}\tilde{g}_{(x,u)}] + b^\top B^\top G_{(x,u)} B b.$$

Define $\mu_{(x,u)} := \mathbb{E}_{(x,u)}[\tilde{X}_{(x,u)}\tilde{g}_{(x,u)}] \in \mathbb{R}^p$ and $H := B^\top G_{(x,u)} B \in \mathbb{R}^{d \times d}$. Then:

$$L(b) = \mathbb{E}_{(x,u)}[\tilde{g}_{(x,u)}^2] - 2b^\top B^\top \mu_{(x,u)} + b^\top H b. \tag{25}$$

*Step 2.2: Solve the first-order condition.* Setting $\nabla_b L(b) = 0$:

$$B^\top G_{(x,u)} B \cdot b = B^\top \mu_{(x,u)}.$$

We verify that $H = B^\top G_{(x,u)} B$ is invertible. By Assumption B.2, $G_{(x,u)} \succeq \lambda u^2 I_p$. Since $B \in \mathrm{St}(p,d)$ has orthonormal columns, for any $v \in \mathbb{R}^d$:

$$v^\top H v = v^\top B^\top G_{(x,u)} B v = (Bv)^\top G_{(x,u)} (Bv) \geq \lambda u^2 \|Bv\|_2^2 = \lambda u^2 \|v\|_2^2,$$

where the last equality uses $B^\top B = I_d$. Hence $H \succeq \lambda u^2 I_d$, guaranteeing invertibility.

The unique solution is:

$$b_{(x,u)} = (B^\top G_{(x,u)} B)^{-1} B^\top \mathbb{E}_{(x,u)}[\tilde{X}_{(x,u)} \tilde{g}_{(x,u)}]. \tag{26}$$

*Step 3: Quadratic expansion.* Define the linearization residual:

$$L^*_{(x,u)} := \tilde{g}_{(x,u)} - \nabla g(x)^\top \tilde{X}_{(x,u)}. \tag{27}$$

We can write:

$$\tilde{g}_{(x,u)} - b_{(x,u)}^\top B^\top \tilde{X}_{(x,u)} = L^*_{(x,u)} + (\nabla g(x) - Bb_{(x,u)})^\top \tilde{X}_{(x,u)}.$$

Expanding the square:

$$\mathbb{E}_{(x,u)} \left[ (\tilde{g}_{(x,u)} - b_{(x,u)}^\top B^\top \tilde{X}_{(x,u)})^2 \right] \tag{28}$$

$$= \mathbb{E}_{(x,u)}[(L^*_{(x,u)})^2] + 2\mathbb{E}_{(x,u)} \left[ L^*_{(x,u)} \cdot (\nabla g(x) - Bb_{(x,u)})^\top \tilde{X}_{(x,u)} \right] + \|\nabla g(x) - Bb_{(x,u)}\|^2_{G_{(x,u)}}, \tag{29}$$

where $\|v\|^2_{G_{(x,u)}} := v^\top G_{(x,u)} v$.

*Step 4: Bound the first term.* By Assumption B.1:

$$\mathbb{E}_{(x,u)}[(L^*_{(x,u)})^2] \leq \frac{L^2}{4} u^4. \tag{30}$$

*Step 5: Bound the cross term.* Let $v := \nabla g(x) - Bb_{(x,u)}$. By the Cauchy–Schwarz inequality:

$$\left| \mathbb{E}_{(x,u)}[L^*_{(x,u)} \cdot v^\top \tilde{X}_{(x,u)}] \right| \leq \sqrt{\mathbb{E}_{(x,u)}[(L^*_{(x,u)})^2]} \cdot \sqrt{\mathbb{E}_{(x,u)}[(v^\top \tilde{X}_{(x,u)})^2]}.$$

For the first factor, by (30):

$$\sqrt{\mathbb{E}_{(x,u)}[(L^*_{(x,u)})^2]} \leq \frac{L}{2} u^2.$$

For the second factor, note that $\mathbb{E}_{(x,u)}[\|\tilde{X}_{(x,u)}\|_2^2] \leq \mathbb{E}_{(x,u)}[\|X - x\|_2^2] \leq u^2$ (since $X \in B(x,u)$ under $\mathbb{E}_{(x,u)}$, and by the variance-minimization property: the covariance is bounded by the second moment about any fixed point). Thus:

$$\mathbb{E}_{(x,u)}[(v^\top \tilde{X}_{(x,u)})^2] \leq \|v\|_2^2 \cdot \mathbb{E}_{(x,u)}[\|\tilde{X}_{(x,u)}\|_2^2] \leq u^2 \|v\|_2^2.$$

Combining:

$$\left| \mathbb{E}_{(x,u)}[L^*_{(x,u)} \cdot v^\top \tilde{X}_{(x,u)}] \right| \leq \frac{L}{2} u^2 \cdot u \|v\|_2 = \frac{L}{2} u^3 \|v\|_2. \tag{31}$$

*Step 6: Lower bound on the quadratic term.* Since $Bb_{(x,u)} \in \mathrm{span}(B)$, by the Pythagorean theorem:

$$\|\nabla g(x) - Bb_{(x,u)}\|_2^2 = \|(I_p - P_B)\nabla g(x)\|_2^2 + \|P_B \nabla g(x) - Bb_{(x,u)}\|_2^2 \geq \|(I_p - P_B)\nabla g(x)\|_2^2. \tag{32}$$

By Assumption B.2:

$$\|\nabla g(x) - Bb_{(x,u)}\|^2_{G_{(x,u)}} \geq \lambda u^2 \|\nabla g(x) - Bb_{(x,u)}\|_2^2 \geq \lambda u^2 \|(I_p - P_B)\nabla g(x)\|_2^2. \tag{33}$$

*Step 7: Combine the bounds.* Let $\delta := \|(I_p - P_B)\nabla g(x)\|_2$. From (28), dropping the non-negative term $\mathbb{E}_{(x,u)}[(L^*_{(x,u)})^2]$ and using (31) and (33):

$$\mathbb{E}_{(x,u)}\left[(\tilde{g}_{(x,u)} - b^\top_{(x,u)}B^\top\tilde{X}_{(x,u)})^2\right] \geq \lambda u^2\|v\|_2^2 - 2\cdot\frac{L}{2}u^3\|v\|_2$$
$$= \lambda u^2\|v\|_2^2 - Lu^3\|v\|_2$$
$$= u^2\|v\|_2\left(\lambda\|v\|_2 - Lu\right).$$

Since $\|v\|_2 \geq \delta$ by (32), whenever $\lambda\delta \geq Lu$ (equivalently, $uL \leq \lambda\delta$), we have:

$$\lambda\|v\|_2 - Lu \geq \lambda\delta - Lu \geq 0.$$

To obtain the coefficient $\lambda/2$ in the final bound, we need:

$$\lambda u^2\|v\|_2^2 - Lu^3\|v\|_2 \geq \frac{\lambda u^2}{2}\|v\|_2^2.$$

This simplifies to:

$$\frac{\lambda u^2}{2}\|v\|_2 \geq Lu^3 \quad\Longleftrightarrow\quad \|v\|_2 \geq \frac{2L}{\lambda}u.$$

Since $\|v\|_2 \geq \delta$, a sufficient condition is $\delta \geq \frac{2L}{\lambda}u$, i.e.,

$$uL \leq \frac{\lambda}{2}\delta = \frac{\lambda}{2}\|(I_p - P_B)\nabla g(x)\|_2.$$

Under this condition:

$$\mathbb{E}_{(x,u)}\left[(\tilde{g}_{(x,u)} - b^\top_{(x,u)}B^\top\tilde{X}_{(x,u)})^2\right] \geq \frac{\lambda u^2}{2}\|v\|_2^2 \geq \frac{\lambda u^2}{2}\delta^2 = \frac{\lambda u^2}{2}\|(I_p - P_B)\nabla g(x)\|_2^2.$$

Combining with the bias-variance decomposition (19):

$$D_{(x,u)}(B) \geq \mathbb{E}_{(x,u)}[\sigma^2(X)] + \frac{\lambda u^2}{2}\|(I_p - P_B)\nabla g(x)\|_2^2.$$

$\square$

## B.2. Proof of Proposition 2.5

Let $\tilde{\mathbf{B}}$ be a minimizer of $D_u(\mathbf{B}) = \int D_{(x,u)}(\mathbf{B})\,\mathbb{P}(\mathrm{d}x)$. Since the central mean subspace has dimension at most $d$, there exists $\mathbf{B}_* \in \mathrm{St}(p,d)$ such that $\nabla g(x) \in \mathrm{span}(\mathbf{B}_*)$ for $\mathbb{P}$-almost every $x$. Define $\tilde{w}(x) := (I_p - P_{\tilde{\mathbf{B}}})\nabla g(x)$.

*Step 1: Upper bound on $D_u(\tilde{\mathbf{B}})$.* By optimality of $\tilde{\mathbf{B}}$ and Proposition 2.4(ii) applied to $\mathbf{B}_*$:

$$D_u(\tilde{\mathbf{B}}) \leq D_u(\mathbf{B}_*) \leq \int \mathbb{E}_{(x,u)}[\sigma^2(X)]\,\mathbb{P}(\mathrm{d}x) + \frac{L^2u^4}{4}.$$

*Step 2: Lower bound on $D_u(\tilde{\mathbf{B}})$.* Partition the support according to whether the condition in Proposition 2.4(i) holds:

$$A = \left\{x : \|\tilde{w}(x)\|_2 \geq \frac{2Lu}{\lambda}\right\}, \qquad A^c = \left\{x : \|\tilde{w}(x)\|_2 < \frac{2Lu}{\lambda}\right\}.$$

On $A$, we have $u \leq \frac{\lambda}{2L}\|\tilde{w}(x)\|_2$, so Proposition 2.4(i) yields

$$D_{(x,u)}(\tilde{\mathbf{B}}) \geq \mathbb{E}_{(x,u)}[\sigma^2(X)] + \frac{\lambda u^2}{2}\|\tilde{w}(x)\|_2^2.$$

On $A^c$, we use the trivial bound $D_{(x,u)}(\tilde{\mathbf{B}}) \geq \mathbb{E}_{(x,u)}[\sigma^2(X)]$. Integrating over each region:

$$D_u(\tilde{\mathbf{B}}) = \int_A D_{(x,u)}(\tilde{\mathbf{B}})\,\mathbb{P}(\mathrm{d}x) + \int_{A^c} D_{(x,u)}(\tilde{\mathbf{B}})\,\mathbb{P}(\mathrm{d}x) \geq \int \mathbb{E}_{(x,u)}[\sigma^2(X)]\,\mathbb{P}(\mathrm{d}x) + \frac{\lambda u^2}{2}\int_A \|\tilde{w}(x)\|_2^2\,\mathbb{P}(\mathrm{d}x).$$

*Step 3: Combine bounds.* Comparing Steps 1 and 2:

$$\frac{\lambda u^2}{2}\int_A \|\tilde{w}(x)\|_2^2\,\mathbb{P}(\mathrm{d}x) \leq \frac{L^2 u^4}{4},$$

which gives $\int_A \|\tilde{w}(x)\|_2^2\,\mathbb{P}(\mathrm{d}x) \leq \frac{L^2 u^2}{2\lambda}$. On $A^c$, by definition $\|\tilde{w}(x)\|_2^2 < \frac{4L^2 u^2}{\lambda^2}$, so

$$\int_{A^c} \|\tilde{w}(x)\|_2^2\,\mathbb{P}(\mathrm{d}x) \leq \frac{4L^2 u^2}{\lambda^2}\mathbb{P}(A^c) \leq \frac{4L^2 u^2}{\lambda^2}.$$

*Step 4: Conclude.* Adding the contributions and using $\lambda \leq 1$:

$$E(\tilde{\mathbf{B}}, \mathcal{M}) = \int_A \|\tilde{w}(x)\|_2^2\,\mathbb{P}(\mathrm{d}x) + \int_{A^c} \|\tilde{w}(x)\|_2^2\,\mathbb{P}(\mathrm{d}x) \leq \frac{L^2 u^2}{2\lambda} + \frac{4L^2 u^2}{\lambda^2} \leq \frac{9L^2 u^2}{2\lambda^2}.$$

## B.3. Proof of Proposition 2.7

Introduce the following notation. For each $j \in \{1, \ldots, n\}$, define: the weighted means $\bar{Y}^{(j)} = \sum_{i=1}^n w_{ij} Y_i$ and $\bar{X}^{(j)} = \sum_{i=1}^n w_{ij} X_i$, and the weight matrix $W^{(j)} = \mathrm{diag}(w_{1j}, \ldots, w_{nj}) \in \mathbb{R}^{n \times n}$. The proof is based on the following lemma.

**Lemma B.3** (Local regression subproblem). *For fixed $\mathbf{B} \in \mathrm{St}(p, d)$ and $j \in \{1, \ldots, n\}$, the local regression coefficients solving*

$$\min_{a,\mathbf{b}} \sum_{i=1}^n [Y_i - a - \mathbf{b}^\top \mathbf{B}^\top (X_i - X_j)]^2 w_{ij}$$

*are given by*

$$\hat{a}_j = \bar{Y}^{(j)} - \hat{\mathbf{b}}_j^\top \mathbf{B}^\top (\bar{X}^{(j)} - X_j),$$
$$\hat{\mathbf{b}}_j = (\mathbf{B}^\top G^{(j)} \mathbf{B})^* \mathbf{B}^\top \mu^{(j)},$$

*where $(\cdot)^*$ denotes the Moore-Penrose pseudoinverse. Moreover, the corresponding residual sum of squares equals*

$$\sigma_j^2(\mathbf{B}) := \sum_{i=1}^n [Y_i - \hat{a}_j - \hat{\mathbf{b}}_j^\top \mathbf{B}^\top (X_i - X_j)]^2 w_{ij} = \|(I_n - H^{(j)}) W^{(j)1/2} \tilde{Y}^{(j)}\|_2^2,$$

*where $H^{(j)} = W^{(j)1/2} \tilde{Z}^{(j)} \mathbf{B} \left(\mathbf{B}^\top G^{(j)} \mathbf{B}\right)^* \mathbf{B}^\top (\tilde{Z}^{(j)})^\top W^{(j)1/2}$ is the orthogonal projection onto the column space of $W^{(j)1/2} \tilde{Z}^{(j)} \mathbf{B}$.*

*Proof.* Setting $\partial_a D_{n,j} = 0$ where $D_{n,j}(\mathbf{B}, a, \mathbf{b}) = \sum_{i=1}^n [Y_i - a - \mathbf{b}^\top \mathbf{B}^\top (X_i - X_j)]^2 w_{ij}$, and using $\sum_i w_{ij} = 1$, yields

$$a = \bar{Y}^{(j)} - \mathbf{b}^\top \mathbf{B}^\top (\bar{X}^{(j)} - X_j).$$

Substituting back, the problem reduces to the centered weighted least squares

$$\min_{\mathbf{b}} \sum_{i=1}^n \left\{\tilde{Y}_i^{(j)} - \mathbf{b}^\top \mathbf{B}^\top \tilde{X}_i^{(j)}\right\}^2 w_{ij} = \min_{\mathbf{b}} \|W^{(j)1/2}(\tilde{Y}^{(j)} - \tilde{Z}^{(j)} \mathbf{B} \mathbf{b})\|_2^2,$$

which is a weighted least squares problem with design matrix $\tilde{Z}^{(j)} \mathbf{B} \in \mathbb{R}^{n \times d}$. The normal equations give the stated formula for $\hat{\mathbf{b}}_j$, and the minimum value is the squared norm of the residual after projection. $\qquad\square$

By Lemma B.3, after profiling out the local regression coefficients $(a_j, \mathbf{b}_j)$, the criterion (5) becomes

$$\min_{\mathbf{B} \in \mathrm{St}(p,d)} \sum_{j=1}^{n} \sigma_j^2(\mathbf{B}) = \min_{\mathbf{B} \in \mathrm{St}(p,d)} \sum_{j=1}^{n} \|(I_n - H^{(j)})W^{(j)1/2}\tilde{Y}^{(j)}\|_2^2.$$

Since $H^{(j)}$ is an orthogonal projection, we have the decomposition

$$\|W^{(j)1/2}\tilde{Y}^{(j)}\|_2^2 = \|(I_n - H^{(j)})W^{(j)1/2}\tilde{Y}^{(j)}\|_2^2 + \|H^{(j)}W^{(j)1/2}\tilde{Y}^{(j)}\|_2^2.$$

Since the first term on the left-hand side is constant with respect to $\mathbf{B}$, minimizing the residual sum of squares is equivalent to maximizing $\sum_{j=1}^{n} \|H^{(j)}W^{(j)1/2}\tilde{Y}^{(j)}\|_2^2$. Finally, a direct computation shows

$$\|H^{(j)}W^{(j)1/2}\tilde{Y}^{(j)}\|_2^2 = (\mu^{(j)})^\top \mathbf{B}(\mathbf{B}^\top G^{(j)}\mathbf{B})^* \mathbf{B}^\top \mu^{(j)},$$

which yields the result. For any orthogonal $\mathbf{Q}$,

$$(\mathbf{BQ})^\top G^{(j)}(\mathbf{BQ}) = \mathbf{Q}^\top(\mathbf{B}^\top G^{(j)}\mathbf{B})\mathbf{Q},$$

and since the pseudoinverse satisfies $(\mathbf{Q}^\top \mathbf{M}\mathbf{Q})^* = \mathbf{Q}^\top \mathbf{M}^*\mathbf{Q}$ for orthogonal $\mathbf{Q}$, we have

$$(\mathbf{BQ})^\top \mu^{(j)} \left[(\mathbf{BQ})^\top G^{(j)}(\mathbf{BQ})\right]^* (\mathbf{BQ})^\top \mu^{(j)}$$
$$= (\mu^{(j)})^\top \mathbf{B} \left(\mathbf{B}^\top G^{(j)}\mathbf{B}\right)^* \mathbf{B}^\top \mu^{(j)}.$$

## C. Proof of Section 3: Equivalence Under Isotropy of MAVE and OPG

**Proposition C.1** (Equivalence OPG with MAVE). *When $G^{(j)} = c \cdot I_p$ for all $j$ and some constant $c > 0$, the MAVE and OPG estimators coincide. Both yield the top $d$ eigenvectors of $M := \sum_{j=1}^{n} \mu^{(j)}(\mu^{(j)})^\top$.*

*Proof.* Under the assumption $G^{(j)} = c \cdot I_p$, the OPG gradient estimates simplify to $\hat{b}_j = c^{-1}\mu^{(j)}$, so

$$\hat{\Sigma}_{\mathrm{OPG}} = \frac{1}{nc^2} \sum_{j=1}^{n} \mu^{(j)}(\mu^{(j)})^\top = \frac{1}{nc^2}M.$$

Thus, $\hat{\mathbf{B}}_{\mathrm{OPG}}$ consists of the top $d$ eigenvectors of $M$.

For MAVE, since $\mathbf{B}^\top G^{(j)}\mathbf{B} = c \cdot \mathbf{B}^\top \mathbf{B} = c \cdot I_d$ for $\mathbf{B} \in \mathrm{St}(p,d)$, the objective (6) becomes

$$F(\mathbf{B}) = \frac{1}{c} \sum_{j=1}^{n} \left\|\mathbf{B}^\top \mu^{(j)}\right\|^2 = \frac{1}{c} \mathrm{tr}\left(\mathbf{B}^\top M\mathbf{B}\right).$$

This is maximized by the top $d$ eigenvectors of $M$.

$\square$

## D. Proof of Section 4: SMAVE Algorithm

### D.1. Proof of Proposition 4.1

We derive the Riemannian gradient of the MAVE objective (6). We first establish the gradient for a single summand.

**Lemma D.1.** *Let $G \in \mathbb{R}^{p \times p}$ be symmetric positive semi-definite and $\mu \in \mathbb{R}^p$. Define $f : \mathrm{St}(p,d) \to \mathbb{R}$ by $f(\mathbf{B}) = \mu^\top \mathbf{B}(\mathbf{B}^\top G\mathbf{B})^{-1}\mathbf{B}^\top \mu$, assuming $\mathbf{B}^\top G\mathbf{B}$ is invertible. Then:*

*1. The Euclidean gradient is $\nabla f(\mathbf{B}) = 2(\mu - G\mathbf{B}u)u^\top$, where $u = (\mathbf{B}^\top G\mathbf{B})^{-1}\mathbf{B}^\top \mu$.*

2. *The Euclidean gradient already lies in the tangent space:* $\nabla f(\mathbf{B}) \in T_{\mathbf{B}}\mathrm{St}(p, d)$.

3. *Consequently,* $\mathrm{grad}\, f(\mathbf{B}) = \nabla f(\mathbf{B})$.

*Proof.* Let $M = \mathbf{B}^\top G\mathbf{B}$, $v = \mathbf{B}^\top \mu$, and $u = M^{-1}v$, so that $f(\mathbf{B}) = v^\top u$.

*Part (i).* Consider a perturbation $\Delta \in \mathbb{R}^{p\times d}$. The differentials are $dv = \Delta^\top \mu$ and $dM = \Delta^\top G\mathbf{B} + \mathbf{B}^\top G\Delta$. Using $d(M^{-1}) = -M^{-1}(dM)M^{-1}$, we compute

$$
\begin{aligned}
df &= 2(dv)^\top u + v^\top d(M^{-1})v \\
&= 2\mu^\top \Delta u - u^\top(\Delta^\top G\mathbf{B} + \mathbf{B}^\top G\Delta)u \\
&= 2\mu^\top \Delta u - 2u^\top \mathbf{B}^\top G\Delta u \\
&= 2\,\mathrm{tr}\big(u(\mu - G\mathbf{B}u)^\top \Delta\big) \\
&= \langle 2(\mu - G\mathbf{B}u)u^\top, \Delta\rangle.
\end{aligned}
$$

Hence $\nabla f(\mathbf{B}) = 2(\mu - G\mathbf{B}u)u^\top$.

*Part (ii).* To verify that $\nabla f(\mathbf{B}) \in T_{\mathbf{B}}\mathrm{St}(p, d)$, we check that $\mathbf{B}^\top \nabla f(\mathbf{B})$ is skew-symmetric:

$$
\mathbf{B}^\top \nabla f(\mathbf{B}) = 2(\mathbf{B}^\top \mu - \mathbf{B}^\top G\mathbf{B}u)u^\top = 2(v - Mu)u^\top = 0,
$$

since $Mu = v$ by definition of $u$. A zero matrix is trivially skew-symmetric.

*Part (iii).* Since $\nabla f(\mathbf{B}) \in T_{\mathbf{B}}\mathrm{St}(p, d)$, the projection is the identity: $\mathrm{grad}\, f(\mathbf{B}) = \Pi_{T_{\mathbf{B}}\mathrm{St}}(\nabla f(\mathbf{B})) = \nabla f(\mathbf{B})$. $\qquad\square$

Proposition 4.1 follows immediately by summing the contributions from each $j \in \{1, \dots, n\}$:

$$
\mathrm{grad}\, F(\mathbf{B}) = \sum_{j=1}^n \mathrm{grad}\, f_j(\mathbf{B}) = 2\sum_{j=1}^n \left(\mu^{(j)} - G^{(j)}\mathbf{B}u^{(j)}\right)(u^{(j)})^\top.
$$

*Remark* D.2 (Horizontality). The identity $\mathbf{B}^\top \nabla f(\mathbf{B}) = 0$ is strictly stronger than tangency: $\nabla f(\mathbf{B})$ lies in the *horizontal* space of the Riemannian submersion $\mathrm{St}(p, d) \to \mathrm{Gr}(p, d)$. This reflects the $\mathcal{O}(d)$-invariance of $f$ (and hence of $F_\varepsilon$): the objective descends to a function on the Grassmannian, and its Stiefel gradient automatically points in directions transverse to the fiber.

*Remark* D.3 (Computational simplification). Because $\nabla F(\mathbf{B})$ already lies in $T_{\mathbf{B}}\mathrm{St}(p, d)$, no explicit tangent-space projection is needed, which reduces per-iteration cost and simplifies implementation. This identification of Riemannian and Euclidean gradients is specific to the embedded Frobenius metric: under the canonical metric of Edelman et al. (1998) or preconditioned metrics such as the one induced by $\bar{G} := n^{-1}\sum_{j=1}^n G^{(j)}$, an explicit metric tensor must be inverted.

*Remark* D.4 (Regularization for singular cases). When $\mathbf{B}^\top G^{(j)}\mathbf{B}$ is singular or ill-conditioned (for instance when $p > k - 1$, in which case $G^{(j)}$ has rank at most $k - 1 < p$) we consider the regularized objective

$$
F_\varepsilon(\mathbf{B}) = \sum_{j=1}^n (\mu^{(j)})^\top \mathbf{B}\big(\mathbf{B}^\top G_\varepsilon^{(j)}\mathbf{B}\big)^{-1}\mathbf{B}^\top \mu^{(j)}, \qquad G_\varepsilon^{(j)} := G^{(j)} + \varepsilon\, I_p, \qquad \varepsilon > 0.
$$

Proposition 4.1 applies verbatim to $F_\varepsilon$ since $\mathbf{B}^\top G_\varepsilon^{(j)}\mathbf{B} = \mathbf{B}^\top G^{(j)}\mathbf{B} + \varepsilon I_d \succeq \varepsilon I_d$, hence the inverse is well-defined for every $\mathbf{B} \in \mathrm{St}(p, d)$. As a result, the Riemannian gradient of $F_\varepsilon$ takes the same closed form, which we encode as a sum of *per-sample terms*

$$
\mathrm{grad}\, F_\varepsilon(\mathbf{B}) = \sum_{j=1}^n h_{j,\varepsilon}(\mathbf{B}), \qquad h_{j,\varepsilon}(\mathbf{B}) := 2\big(\mu^{(j)} - G_\varepsilon^{(j)}\mathbf{B}\, u_\varepsilon^{(j)}\big)(u_\varepsilon^{(j)})^\top, \qquad u_\varepsilon^{(j)} := \big(\mathbf{B}^\top G_\varepsilon^{(j)}\mathbf{B}\big)^{-1}\mathbf{B}^\top \mu^{(j)},
$$

with $h_{j,\varepsilon}(\mathbf{B}) \in T_{\mathbf{B}}\mathrm{St}(p, d)$ without projection. Note that considering the regularized objective $F_\varepsilon$ corresponds to profiling out *ridge-regularized* local regression coefficients:

$$
\widehat{\mathbf{b}}_j^\varepsilon = \arg\min_{\mathbf{b}\in\mathbb{R}^d} \left\{ \sum_{i=1}^n \big(Y_i - a - \mathbf{b}^\top \mathbf{B}^\top(X_i - X_j)\big)^2 w_{ij} + \varepsilon\|\mathbf{b}\|^2 \right\}.
$$

**D.2. Convergence analysis: setup and key estimates**

We work in the simplified setting of Section 4.3: fixed neighborhoods, no momentum, no gradient normalization, mini-batch size $m \geq 1$ drawn uniformly without replacement from $\{1, \ldots, n\}$, independently across iterations. Data $\{(\mu^{(j)}, G^{(j)})\}_{j=1}^{n}$ are fixed; randomness comes from $\{J_t\}$, generating the filtration $\mathcal{F}_t := \sigma(\mathbf{B}_0, J_0, \ldots, J_{t-1})$. The iterates are

$$\mathbf{B}_{t+1} = R_{\mathbf{B}_t}^{\mathrm{QR}}(\alpha_t \mathbf{g}_t), \qquad \mathbf{g}_t := \frac{n}{m} \sum_{j \in J_t} h_{j,\varepsilon}(\mathbf{B}_t), \tag{34}$$

with $h_{j,\varepsilon}, u_{\varepsilon}^{(j)}$ as in Remark D.4, giving $\mathbf{g}_t \in T_{\mathbf{B}_t}\mathrm{St}(p,d)$ a.s. Since $\mathbf{g}_t = \phi(\mathbf{B}_t, J_t)$ with $J_t \perp\!\!\!\perp \mathcal{F}_t$ and $\mathbf{B}_t$ being $\mathcal{F}_t$-measurable, the standard freezing identity (Kallenberg, 1997) gives $\mathbb{E}[\mathbf{g}_t \mid \mathcal{F}_t] = \mathbb{E}[\mathbf{g}_t \mid \mathbf{B}_t]$, and likewise for $\|\mathbf{g}_t\|_F^2$. Uniform sampling yields $\Pr(j \in J_t \mid \mathcal{F}_t) = m/n$, hence $\mathbb{E}[\mathbf{g}_t \mid \mathcal{F}_t] = \mathrm{grad}\, F_{\varepsilon}(\mathbf{B}_t)$.

Define $\mathcal{U} := \{\mathbf{Z} \in \mathbb{R}^{p \times d} : \mathbf{Z}^\top G_{\varepsilon}^{(j)} \mathbf{Z} \succ 0 \; \forall j\}$; this is open and contains $\mathrm{St}(p,d)$ since $v^\top \mathbf{B}^\top G_{\varepsilon}^{(j)} \mathbf{B} v \geq \varepsilon \|v\|_2^2$ for $\mathbf{B} \in \mathrm{St}(p,d)$. As $A \mapsto A^{-1}$ is real-analytic on $\mathrm{GL}_d(\mathbb{R})$, $F_{\varepsilon}$ extends to a $C^\infty$ function on $\mathcal{U}$. Define the data-dependent constants

$$M_\mu := \max_j \|\mu^{(j)}\|_2, \quad M_G := \max_j \|G_{\varepsilon}^{(j)}\|_{\mathrm{op}}, \quad \lambda_\varepsilon := \min_j \lambda_{\min}(G_{\varepsilon}^{(j)}).$$

Each of $M_\mu, M_G$ is finite for any finite sample, and $\lambda_\varepsilon \geq \varepsilon$ by construction of $G_{\varepsilon}^{(j)} = G^{(j)} + \varepsilon I_p$, so these are not assumptions but notational shorthand. All constants below depend only on $(n, d, p, M_\mu, M_G, \lambda_\varepsilon)$ and a $(p,d)$-dependent QR-retraction constant $c_\mathrm{R}$ (Absil & Malick, 2012). We write $C := -F_\varepsilon$ to apply standard *minimization* results.

**Sign and boundedness of $F_\varepsilon$.** For every $\mathbf{B} \in \mathrm{St}(p,d)$ and every $j$, the matrix $\mathbf{B}^\top G_{\varepsilon}^{(j)} \mathbf{B} \succeq \varepsilon I_d \succ 0$ is invertible with a positive-definite inverse, so each summand of $F_\varepsilon$ is a non-negative quadratic form in $\mathbf{B}^\top \mu^{(j)}$. Hence

$$0 \leq F_\varepsilon(\mathbf{B}) \leq \sum_{j=1}^{n} \frac{\|\mathbf{B}^\top \mu^{(j)}\|_2^2}{\lambda_\varepsilon} \leq \frac{nM_\mu^2}{\lambda_\varepsilon}, \qquad \forall \mathbf{B} \in \mathrm{St}(p,d). \tag{35}$$

Let

$$\Delta_\varepsilon := \sup_{\mathrm{St}(p,d)} F_\varepsilon - \inf_{\mathrm{St}(p,d)} F_\varepsilon \leq \sup_{\mathrm{St}(p,d)} F_\varepsilon \leq \frac{nM_\mu^2}{\lambda_\varepsilon} \tag{36}$$

by (35); this bound is deterministic and independent of $\mathbf{B}_0$.

**Uniform pointwise bounds.** Set $A_0 := 2M_\mu^2(1 + M_G/\lambda_\varepsilon)/\lambda_\varepsilon$. For $\mathbf{B} \in \mathrm{St}(p,d)$, $\|\mathbf{B}\|_{\mathrm{op}} = 1$ and $\|(\mathbf{B}^\top G_{\varepsilon}^{(j)} \mathbf{B})^{-1}\|_{\mathrm{op}} \leq 1/\lambda_\varepsilon$, hence $\|u_{\varepsilon}^{(j)}\|_2 \leq M_\mu/\lambda_\varepsilon$. The rank-one identity $\|ab^\top\|_F = \|a\|_2 \|b\|_2$ then yields, for every $\mathbf{B} \in \mathrm{St}(p,d)$ and every $j$,

$$\|h_{j,\varepsilon}(\mathbf{B})\|_F \leq A_0, \qquad \|\mathrm{grad}\, F_\varepsilon(\mathbf{B})\|_F \leq nA_0, \qquad \|\mathbf{g}_t\|_F \leq nA_0 \text{ pointwise.} \tag{37}$$

**Retraction smoothness.** We use the QR quadratic-error bound: there exist $\rho_0 \in (0,1)$ and $c_\mathrm{R} = c_\mathrm{R}(p,d) > 0$ such that

$$\left\| R_{\mathbf{B}}^{\mathrm{QR}}(\xi) - \mathbf{B} - \xi \right\|_F \leq c_\mathrm{R} \|\xi\|_F^2, \qquad \mathbf{B} \in \mathrm{St}(p,d), \; \|\xi\|_F \leq \rho_0, \tag{38}$$

which follows from Taylor expansion of the $C^\infty$ QR retraction on the compact $\mathrm{St}(p,d)$ (Boumal, 2023).

**Lemma D.5** (Retraction smoothness of $F_\varepsilon$). *There exist a radius $\rho = \rho(p,d) > 0$ and a constant*

$$L_\varepsilon = 2c_\mathrm{R}\, nA_0 + \frac{9}{4} L_\nabla, \qquad L_\nabla \leq \frac{C_* nM_\mu^2(1 + M_G/\lambda_\varepsilon)^2}{\lambda_\varepsilon}, \tag{39}$$

*with $C_* > 0$ absolute, such that $C = -F_\varepsilon$ is retraction $L_\varepsilon$-smooth on $\mathrm{St}(p,d)$ w.r.t. the QR retraction:*

$$C\left(R_{\mathbf{B}}^{\mathrm{QR}}(\xi)\right) \leq C(\mathbf{B}) + \langle \mathrm{grad}\, C(\mathbf{B}), \xi \rangle + \tfrac{L_\varepsilon}{2}\|\xi\|_F^2, \qquad \forall \mathbf{B} \in \mathrm{St}(p,d), \; \|\xi\|_F \leq \rho. \tag{40}$$

*In particular $L_\varepsilon = \mathcal{O}\big(nM_\mu^2(1 + M_G/\lambda_\varepsilon)^2/\lambda_\varepsilon\big)$ as $\varepsilon \to 0$, with the implied constant depending only on $(p,d)$.*

*Proof.* We prove the equivalent lower retraction smoothness for $F_\varepsilon$. For $\mathbf{Z} \in \mathbb{R}^{p \times d}$, $\sigma_{\min}(\mathbf{Z}) := \sqrt{\lambda_{\min}(\mathbf{Z}^\top \mathbf{Z})}$ denotes its smallest singular value. Choose $\rho := \min\{\rho_0,\, 1/(2(1 + c_{\mathrm{R}}))\}$, so that $c_{\mathrm{R}}\rho \leq \frac{1}{2}$ and $(1 + c_{\mathrm{R}}\rho)\rho \leq 3/4$. Define

$$\mathcal{V} := \left\{\mathbf{Z} \in \mathbb{R}^{p \times d} : \sigma_{\min}(\mathbf{Z}) \geq \tfrac{1}{4},\ \|\mathbf{Z}\|_{\mathrm{op}} \leq \tfrac{7}{4}\right\}.$$

Both $\sigma_{\min}$ and $\|\cdot\|_{\mathrm{op}}$ are continuous on $\mathbb{R}^{p \times d}$, so $\mathcal{V}$ is closed; it is bounded since $\|\mathbf{Z}\|_F \leq \sqrt{d}\,\|\mathbf{Z}\|_{\mathrm{op}} \leq 7\sqrt{d}/4$ on $\mathcal{V}$. Hence $\mathcal{V}$ is compact. ($\mathcal{V}$ is not convex, since $\{\sigma_{\min} \geq 1/4\}$ is not convex in general; the descent argument below uses only that the segment $[\mathbf{B}, y]$ lies in $\mathcal{V}$, established in Step 0.)

*Step 0 (Chord in $\mathcal{V}$).* Fix $\mathbf{B} \in \mathrm{St}(p, d)$, $\xi \in T_\mathbf{B}\mathrm{St}(p, d)$ with $\|\xi\|_F \leq \rho$, and set $y := R_\mathbf{B}^{\mathrm{QR}}(\xi) \in \mathrm{St}(p, d)$. By (38) and the triangle inequality,

$$\|y - \mathbf{B}\|_F \leq \|\xi\|_F + c_{\mathrm{R}}\|\xi\|_F^2 = (1 + c_{\mathrm{R}}\|\xi\|_F)\|\xi\|_F \leq (1 + c_{\mathrm{R}}\rho)\|\xi\|_F \leq (1 + c_{\mathrm{R}}\rho)\rho.$$

The choice $\rho \leq 1/(2(1 + c_{\mathrm{R}}))$ gives $c_{\mathrm{R}}\rho \leq 1/2$ and $(1 + c_{\mathrm{R}}\rho)\rho \leq 3/(4(1 + c_{\mathrm{R}})) \leq 3/4$. For $z_t := (1 - t)\mathbf{B} + ty$, $t \in [0, 1]$, we have $\|z_t - \mathbf{B}\|_{\mathrm{op}} \leq \|z_t - \mathbf{B}\|_F = t\|y - \mathbf{B}\|_F \leq 3/4$. Since $\mathbf{B}^\top \mathbf{B} = I_d$ gives $\sigma_{\min}(\mathbf{B}) = 1$, Weyl's perturbation inequality for singular values (Horn & Johnson, 2013) yields $\sigma_{\min}(z_t) \geq \sigma_{\min}(\mathbf{B}) - \|z_t - \mathbf{B}\|_{\mathrm{op}} \geq 1/4$, and $\|z_t\|_{\mathrm{op}} \leq 1 + 3/4 = 7/4$, so $z_t \in \mathcal{V}$. Hence $[\mathbf{B}, y] \subset \mathcal{V}$.

*Step 1 (Hessian bound on $\mathcal{V}$).* For every $\mathbf{Z} \in \mathcal{V}$, every $j$, and every $v \in \mathbb{R}^d$,

$$v^\top \mathbf{Z}^\top G_\varepsilon^{(j)} \mathbf{Z}\, v = (\mathbf{Z}v)^\top G_\varepsilon^{(j)}(\mathbf{Z}v) \geq \lambda_\varepsilon \|\mathbf{Z}v\|_2^2 \geq (\lambda_\varepsilon/16)\|v\|_2^2,$$

so $M_j(\mathbf{Z}) := \mathbf{Z}^\top G_\varepsilon^{(j)} \mathbf{Z}$ is invertible on $\mathcal{V}$ with $\|M_j(\mathbf{Z})^{-1}\|_{\mathrm{op}} \leq 16/\lambda_\varepsilon$, and $F_\varepsilon$ is $C^\infty$ on $\mathcal{V}$. Write $u_j(\mathbf{Z}) := M_j(\mathbf{Z})^{-1}\mathbf{Z}^\top \mu^{(j)}$ and $r_j(\mathbf{Z}) := \mu^{(j)} - G_\varepsilon^{(j)}\mathbf{Z}\,u_j(\mathbf{Z})$, so that $\nabla f_j(\mathbf{Z}) = 2\,r_j(\mathbf{Z})\,u_j(\mathbf{Z})^\top$. Pointwise on $\mathcal{V}$,

$$\|u_j(\mathbf{Z})\|_2 \leq \tfrac{16}{\lambda_\varepsilon} \cdot \tfrac{7}{4} \cdot M_\mu = \tfrac{28M_\mu}{\lambda_\varepsilon}, \qquad \|r_j(\mathbf{Z})\|_2 \leq M_\mu\left(1 + \tfrac{49M_G}{\lambda_\varepsilon}\right) \leq 49\,M_\mu\left(1 + M_G/\lambda_\varepsilon\right). \tag{41}$$

Throughout this proof, $C_*$ denotes an absolute constant whose value may change from line to line. Differentiating along $\Delta \in \mathbb{R}^{p \times d}$ via $d(M_j^{-1})[\Delta] = -M_j^{-1}\left(\Delta^\top G_\varepsilon^{(j)}\mathbf{Z} + \mathbf{Z}^\top G_\varepsilon^{(j)}\Delta\right)M_j^{-1}$,

$$du_j[\Delta] = -M_j^{-1}\left(\Delta^\top G_\varepsilon^{(j)}\mathbf{Z} + \mathbf{Z}^\top G_\varepsilon^{(j)}\Delta\right)M_j^{-1}\mathbf{Z}^\top \mu^{(j)} + M_j^{-1}\Delta^\top \mu^{(j)},$$

$$dr_j[\Delta] = -G_\varepsilon^{(j)}\Delta\,u_j - G_\varepsilon^{(j)}\mathbf{Z}\,du_j[\Delta].$$

Substituting $\|M_j^{-1}\|_{\mathrm{op}} \leq 16/\lambda_\varepsilon$, $\|\mathbf{Z}\|_{\mathrm{op}} \leq 7/4$, $\|G_\varepsilon^{(j)}\|_{\mathrm{op}} \leq M_G$, and (41),

$$\|du_j[\Delta]\|_2 \leq \frac{C_* M_\mu(1 + M_G/\lambda_\varepsilon)}{\lambda_\varepsilon}\|\Delta\|_F, \qquad \|dr_j[\Delta]\|_2 \leq \frac{C_* M_G M_\mu(1 + M_G/\lambda_\varepsilon)}{\lambda_\varepsilon}\|\Delta\|_F.$$

By the rank-one Frobenius identity $\|ab^\top\|_F = \|a\|_2\|b\|_2$,

$$\|\nabla^2 f_j(\mathbf{Z})[\Delta]\|_F \leq 2\left(\|dr_j[\Delta]\|_2\|u_j\|_2 + \|r_j\|_2\|du_j[\Delta]\|_2\right).$$

Substituting (41) and the displayed bounds on $\|du_j[\Delta]\|_2, \|dr_j[\Delta]\|_2$ gives two summands:

$$\|dr_j[\Delta]\|_2\|u_j\|_2 \leq \frac{C_* M_G\,M_\mu^2(1 + M_G/\lambda_\varepsilon)}{\lambda_\varepsilon^2}\|\Delta\|_F, \qquad \|r_j\|_2\|du_j[\Delta]\|_2 \leq \frac{C_* M_\mu^2(1 + M_G/\lambda_\varepsilon)^2}{\lambda_\varepsilon}\|\Delta\|_F.$$

Using $M_G/[\lambda_\varepsilon(1 + M_G/\lambda_\varepsilon)] = M_G/(\lambda_\varepsilon + M_G) \leq 1$, the first summand is bounded by the second; hence

$$\|\nabla^2 f_j(\mathbf{Z})[\Delta]\|_F \leq \frac{C_* M_\mu^2(1 + M_G/\lambda_\varepsilon)^2}{\lambda_\varepsilon}\|\Delta\|_F.$$

Summing over $j$ gives the Lipschitz constant of $\nabla F_\varepsilon$ on $\mathcal{V}$:

$$L_\nabla \leq \frac{C_* n M_\mu^2(1 + M_G/\lambda_\varepsilon)^2}{\lambda_\varepsilon}.$$

*Step 2 (Euclidean descent on $[\mathbf{B}, y] \subset \mathcal{V}$).* Since $\nabla F_\varepsilon$ is $L_\nabla$-Lipschitz on the convex segment,

$$F_\varepsilon(y) \geq F_\varepsilon(\mathbf{B}) + \langle \nabla F_\varepsilon(\mathbf{B}), y - \mathbf{B} \rangle - \tfrac{L_\nabla}{2} \|y - \mathbf{B}\|_F^2.$$

*Step 3 (Combine).* By Lemma D.1, $\nabla F_\varepsilon(\mathbf{B}) = \operatorname{grad} F_\varepsilon(\mathbf{B}) \in T_\mathbf{B} \mathrm{St}(p, d)$, so $\langle \nabla F_\varepsilon(\mathbf{B}), \eta \rangle = \langle \operatorname{grad} F_\varepsilon(\mathbf{B}), \eta \rangle$ for any $\eta \in \mathbb{R}^{p \times d}$. Writing $y - \mathbf{B} = \xi + (y - \mathbf{B} - \xi)$, Cauchy–Schwarz with (38) and (37) gives

$$|\langle \operatorname{grad} F_\varepsilon(\mathbf{B}), y - \mathbf{B} - \xi \rangle| \leq n A_0\, c_\mathrm{R} \|\xi\|_F^2$$

and $\|y - \mathbf{B}\|_F \leq (1 + c_\mathrm{R} \rho) \|\xi\|_F \leq \tfrac{3}{2} \|\xi\|_F$, so $\tfrac{L_\nabla}{2} \|y - \mathbf{B}\|_F^2 \leq \tfrac{9 L_\nabla}{8} \|\xi\|_F^2$. Thus

$$F_\varepsilon(y) \geq F_\varepsilon(\mathbf{B}) + \langle \operatorname{grad} F_\varepsilon(\mathbf{B}), \xi \rangle - \underbrace{\left( c_\mathrm{R}\, n A_0 + \tfrac{9}{8} L_\nabla \right)}_{= L_\varepsilon / 2} \|\xi\|_F^2,$$

which is (40), and substituting $A_0 = 2 M_\mu^2 (1 + M_G / \lambda_\varepsilon) / \lambda_\varepsilon$ together with the bound on $L_\nabla$ yields (39). $\qquad\square$

Lemma D.5 is the sole place where the geometry of $\mathrm{St}(p, d)$ and the QR retraction enter the convergence analysis quantitatively; the remaining ingredient is the without-replacement variance bound, which we establish next.

**Without-replacement variance identity.** Applying the scalar without-replacement variance formula (Cochran, 1977) entrywise yields

$$\mathbb{E}\big[\|\mathbf{g}_t - \operatorname{grad} F_\varepsilon(\mathbf{B}_t)\|_F^2 \mid \mathcal{F}_t\big] = \sigma_\varepsilon^2(\mathbf{B}_t) := \frac{n(n - m)}{m(n - 1)} \sum_{j=1}^n \|h_{j,\varepsilon}(\mathbf{B}_t) - \bar{h}_\varepsilon(\mathbf{B}_t)\|_F^2, \tag{42}$$

where $\bar{h}_\varepsilon(\mathbf{B}) := n^{-1} \operatorname{grad} F_\varepsilon(\mathbf{B})$. The bound $\sum_j \|h_{j,\varepsilon} - \bar{h}_\varepsilon\|_F^2 \leq \sum_j \|h_{j,\varepsilon}\|_F^2 \leq n A_0^2$ [from (37)] gives the uniform upper bound

$$\sigma_\varepsilon^2 := \frac{n^2 (n - m) A_0^2}{m(n - 1)}, \qquad \mathbb{E}\big[\|\mathbf{g}_t\|_F^2 \mid \mathcal{F}_t\big] \leq \|\operatorname{grad} F_\varepsilon(\mathbf{B}_t)\|_F^2 + \sigma_\varepsilon^2. \tag{43}$$

In particular $\sigma_\varepsilon^2 = 0$ at $m = n$, and dropping the finite-population factor $(n - m)/[m(n - 1)]$ recovers the crude pointwise bound $\|\mathbf{g}_t\|_F \leq n A_0$ from (37).

### D.3. Proof of Proposition 4.3: asymptotic convergence

We verify the hypotheses of Theorem 2 in Bonnabel (2013) applied to $C = -F_\varepsilon$. The manifold $\mathrm{St}(p, d)$ is connected, compact, and $C^\infty$ embedded in $\mathbb{R}^{p \times d}$, the QR retraction is $C^\infty$ (Absil et al., 2008). The step sizes $\alpha_t = \alpha_0 / (1 + \gamma t)$ satisfy $\sum_t \alpha_t = \infty$ and $\sum_t \alpha_t^2 < \infty$, and the pointwise bound (37) gives the uniform gradient bound $\|\mathbf{g}_t\|_F \leq n A_0$ on $\mathrm{St}(p, d)$.

The remaining hypothesis — a uniform quadratic upper bound on $C \circ R_\mathbf{B}^{\mathrm{QR}}$ along the iterates — is supplied by the retraction-smoothness inequality (40) provided each step lies within its smoothness radius $\rho$. Imposing

$$\alpha_0 \leq \bar{\alpha}_\varepsilon = \min\{1/L_\varepsilon,\ \rho/(n A_0)\} \tag{44}$$

together with (37) yields $\|\alpha_t \mathbf{g}_t\|_F \leq \alpha_0\, n A_0 \leq \rho$ for all $t$, so (40) applies at every iteration with the deterministic constant $L_\varepsilon$. The same bound ensures the QR retraction is well-defined, since $\sigma_{\min}(\mathbf{B} + \alpha_t \mathbf{g}_t) \geq 1 - \rho > 0$. The threshold $\bar{\alpha}_\varepsilon$ in (44) is the "explicit threshold" referenced in the statement of Proposition 4.3.

By Theorem 2 in Bonnabel (2013), $F_\varepsilon(\mathbf{B}_t)$ converges a.s. and $\operatorname{grad} F_\varepsilon(\mathbf{B}_t) \to 0$ a.s., proving (i) and (ii). Conclusion (iii) follows: on the probability-one event from (ii), any accumulation point $\mathbf{B}^\star(\omega)$ of $\{\mathbf{B}_t(\omega)\}$ satisfies $\operatorname{grad} F_\varepsilon(\mathbf{B}^\star(\omega)) = 0$ by continuity of $\operatorname{grad} F_\varepsilon$ on the compact $\mathrm{St}(p, d)$.

### D.4. Proof of Theorem 4.4: non-asymptotic rate

We state and prove the explicit-constant version (Theorem D.6); the main-text bound is immediate from (45)–(46). The two ingredients are the retraction smoothness from Lemma D.5 and the variance identity (42)–(43), both established in the setup.

**Theorem D.6** (Non-asymptotic rate, explicit)**.** *Run* (34) *with constant step size* $\alpha \in (0, \bar{\alpha}_\varepsilon]$*, where* $\bar{\alpha}_\varepsilon :=$ $\min\{1/L_\varepsilon, \rho/(nA_0)\}$*. Then for all* $T \geq 1$,

$$\frac{1}{T}\sum_{t=0}^{T-1} \mathbb{E}\big[\|\mathrm{grad}\, F_\varepsilon(\mathbf{B}_t)\|_F^2\big] \;\leq\; \frac{2\Delta_\varepsilon}{\alpha T} + L_\varepsilon \alpha \sigma_\varepsilon^2. \tag{45}$$

*For* $T \geq T_0 := 2\Delta_\varepsilon/(L_\varepsilon \sigma_\varepsilon^2 \bar{\alpha}_\varepsilon^2)$*, the choice* $\alpha^\star = \sqrt{2\Delta_\varepsilon/(L_\varepsilon \sigma_\varepsilon^2 T)}$ *is admissible and yields*

$$\frac{1}{T}\sum_{t=0}^{T-1} \mathbb{E}\big[\|\mathrm{grad}\, F_\varepsilon(\mathbf{B}_t)\|_F^2\big] \;\leq\; 2\sqrt{\frac{2L_\varepsilon \Delta_\varepsilon \sigma_\varepsilon^2}{T}} \;=\; \frac{2nA_0}{\sqrt{m}}\sqrt{\frac{2L_\varepsilon \Delta_\varepsilon (n-m)}{(n-1)\,T}} \;=\; O\Big(\frac{1}{\sqrt{T}}\Big). \tag{46}$$

*Proof.* The constraint $\alpha \leq \rho/(nA_0)$ and (37) enforce $\|\alpha \mathbf{g}_t\|_F \leq \rho$ pointwise, so Lemma D.5 applies with $\xi = \alpha \mathbf{g}_t$. Taking conditional expectation, using unbiasedness and (43),

$$\mathbb{E}[C(\mathbf{B}_{t+1}) \mid \mathcal{F}_t] \leq C(\mathbf{B}_t) - \alpha\|\mathrm{grad}\, F_\varepsilon(\mathbf{B}_t)\|_F^2 + \tfrac{L_\varepsilon \alpha^2}{2}\big(\|\mathrm{grad}\, F_\varepsilon(\mathbf{B}_t)\|_F^2 + \sigma_\varepsilon^2\big).$$

The constraint $\alpha \leq 1/L_\varepsilon$ gives $\alpha - L_\varepsilon \alpha^2/2 \geq \alpha/2$, yielding the standard non-convex stochastic descent recursion (Boumal et al., 2019):

$$\mathbb{E}[C(\mathbf{B}_{t+1}) \mid \mathcal{F}_t] \leq C(\mathbf{B}_t) - \tfrac{\alpha}{2}\|\mathrm{grad}\, F_\varepsilon(\mathbf{B}_t)\|_F^2 + \tfrac{L_\varepsilon \alpha^2}{2}\sigma_\varepsilon^2. \tag{47}$$

Telescoping (47) and taking full expectations gives

$$
\begin{aligned}
\frac{\alpha}{2}\sum_{t=0}^{T-1}\mathbb{E}\big[\|\mathrm{grad}\, F_\varepsilon(\mathbf{B}_t)\|_F^2\big] &\leq\; \mathbb{E}[C(\mathbf{B}_0)] - \mathbb{E}[C(\mathbf{B}_T)] + T\tfrac{L_\varepsilon \alpha^2}{2}\sigma_\varepsilon^2 \\
&=\; \mathbb{E}[F_\varepsilon(\mathbf{B}_T)] - \mathbb{E}[F_\varepsilon(\mathbf{B}_0)] + T\tfrac{L_\varepsilon \alpha^2}{2}\sigma_\varepsilon^2 \\
&\leq\; \Delta_\varepsilon + T\tfrac{L_\varepsilon \alpha^2}{2}\sigma_\varepsilon^2,
\end{aligned}
$$

where the final inequality uses $\mathbb{E}[F_\varepsilon(\mathbf{B}_T)] \leq \sup_{\mathrm{St}} F_\varepsilon$, $\mathbb{E}[F_\varepsilon(\mathbf{B}_0)] \geq \inf_{\mathrm{St}} F_\varepsilon$, and the definition (36). Dividing by $\alpha T/2$ yields (45). The right-hand side of (45) is of the form $a/\alpha + b\alpha$ with $a = 2\Delta_\varepsilon/T$, $b = L_\varepsilon \sigma_\varepsilon^2$, minimized at $\alpha^\star = \sqrt{a/b}$ with value $2\sqrt{ab}$; the admissibility $\alpha^\star \leq \bar{\alpha}_\varepsilon$ is equivalent to $T \geq T_0$. Substituting (43) gives (46). $\qquad\square$

### D.5. Discussion

**Finite-population correction.** The factor $(n-m)/(n-1) \in [0,1]$ is the standard finite-population correction. At $m = n$, $\sigma_\varepsilon^2 = 0$ and (45) reduces to the deterministic bound $2\Delta_\varepsilon/(\alpha T)$, matching full Riemannian gradient descent.

*Remark* D.7 (Regularization bias)**.** On $\mathcal{B}_c := \{\mathbf{B} \in \mathrm{St}(p,d) : \min_j \lambda_{\min}(\mathbf{B}^\top G^{(j)}\mathbf{B}) \geq c\}$ with $c > 0$, the identity $M^{-1} - (M + \varepsilon I_d)^{-1} = \varepsilon M^{-1}(M + \varepsilon I_d)^{-1}$ applied to $M = \mathbf{B}^\top G^{(j)}\mathbf{B}$ yields

$$0 \;\leq\; F(\mathbf{B}) - F_\varepsilon(\mathbf{B}) \;\leq\; \frac{n\, M_\mu^2\, \varepsilon}{c^2}, \qquad \mathbf{B} \in \mathcal{B}_c.$$

Taking $\varepsilon \ll c$ recovers the unregularized objective uniformly on $\mathcal{B}_c$.

**Behaviour as $\varepsilon \to 0$.** When $p > k - 1$, $G^{(j)}$ is rank-deficient and $\lambda_\varepsilon = \varepsilon$. Substituting into (39)–(43) and using $F_\varepsilon \leq nM_\mu^2/\lambda_\varepsilon$ on $\mathrm{St}(p,d)$ yields $L_\varepsilon = \Theta(n\,\varepsilon^{-3})$, $\sigma_\varepsilon^2 = \Theta(n^2\varepsilon^{-4}/m)$, $\Delta_\varepsilon = \mathcal{O}(n\,\varepsilon^{-1})$ by (36), and $\bar{\alpha}_\varepsilon = \Theta(\varepsilon^3)$ (for $\varepsilon < 1$, the binding constraint is $1/L_\varepsilon = \Theta(\varepsilon^3) < \rho/(nA_0) = \Theta(\varepsilon^2)$). *The admissibility threshold $T_0$ remains $O(1)$ in* $\varepsilon$: $T_0 = 2\Delta_\varepsilon/[L_\varepsilon \sigma_\varepsilon^2 \bar{\alpha}_\varepsilon^2] = O(\varepsilon^{-1}/[\varepsilon^{-3} \cdot \varepsilon^{-4} \cdot \varepsilon^6]) = O(1)$. The overall rate is $\mathcal{O}(n^2\varepsilon^{-4}/\sqrt{mT})$. The degradation is intrinsic since $F_0$ is undefined when $\mathbf{B}^\top G^{(j)}\mathbf{B}$ is rank-deficient. In practice one fixes $\varepsilon \ll \min_j \lambda_{\min}(G^{(j)})$ when the latter is positive, recovering the unregularized minimizer up to the bias of Remark D.7.

## E. Implementation Details

**Kernel and neighborhood-size calibration.** All kernel-based methods (OPG, RMAVE) use the Gaussian kernel $K_h(u) = \exp(-\|u\|^2/2h^2)$ with Silverman's rule-of-thumb bandwidth (Silverman, 2018),

$$h = \hat{\sigma}\, n^{-1/(q+4)}, \tag{48}$$

where $q$ is the operating dimension ($q = p$ for OPG, $q = d$ for RMAVE) and $\hat{\sigma}$ is the median absolute deviation across coordinates, scaled by $1.4826 = 1/\Phi^{-1}(3/4)$. For $k$-NN methods, we match the expected kernel neighborhood size $\sim nh^q$ by setting $k = \lceil n^{4/(q+4)} \rceil$, clipped to $[k_{\min}, n/3]$ with $k_{\min} = 50$ for OPG-$k$NN (operating in $\mathbb{R}^p$) and $k_{\min} = 20$ for RMAVE-$k$NN and SMAVE (operating in $\mathbb{R}^d$).

**Iterative methods.** RMAVE, RMAVE-$k$NN, and SMAVE all run for $T = 100$ iterations. RMAVE and RMAVE-$k$NN are warm-started from the OPG solution, following the original RMAVE protocol (Xia et al., 2002). SMAVE is initialized uniformly on $\mathrm{St}(p, d)$ by orthonormalizing a $p \times d$ matrix with i.i.d. $\mathcal{N}(0, 1)$ entries via QR decomposition. All matrix inversions are regularized with $\varepsilon = 10^{-5}$. Implementations use Python with NumPy/SciPy; random seeds are fixed for reproducibility.

**Runtime accounting.** Reported runtimes are wall-clock seconds. For RMAVE and RMAVE-$k$NN, the timer covers only the refinement loop; the shared OPG warm start is timed once and reported separately under OPG. SMAVE's runtime includes its random Stiefel draw.

**RMAVE-$k$NN.** RMAVE-$k$NN is identical to RMAVE except that the dense Gaussian kernel weights are replaced, at each iteration, by uniform $k$-NN weights in the projected space:

$$w_{ij} = \begin{cases} 1/k & \text{if } X_i \in \mathcal{N}_k(\widehat{\mathbf{B}}, X_j), \\ 0 & \text{otherwise,} \end{cases}$$

with $k = \lceil n^{4/(d+4)} \rceil$ matching SMAVE's default. The coordinate-descent solver, direction update, re-orthogonalization step, OPG initialization, and convergence criterion are unchanged. This isolates the effect of replacing kernel localization with $k$-NN localization in RMAVE's algorithmic structure (see Appendix F.1).

**SMAVE hyperparameters.** We tuned SMAVE hyperparameters via grid search over 7,560 synthetic configurations: five link functions × three sample sizes ($n \in \{1000, 2000, 5000\}$) × two dimensions ($p \in \{10, 20\}$) × five replications × 252 hyperparameter combinations. The tuning grid covered:

- Initial step size: $\alpha_0 \in \{0.05, 0.08, 0.1, 0.12, 0.15, 0.18, 0.2\}$

- Step size decay: $\gamma \in \{0.02, 0.04, 0.06\}$

- Momentum: $\beta \in \{0.0, 0.5, 0.9\}$

- $k$-NN refresh period: $\tau \in \{20, 25, 30, 50\}$

Analysis revealed that momentum hyperparameter $\beta = 0.9$ consistently reduced error by approximately $3\times$ compared to no momentum, making it the most impactful hyperparameter. Larger step sizes ($\alpha_0 \geq 0.15$) improved convergence within the 100-iteration budget. Lower k-NN refresh frequencies (20–30) yielded better accuracy but increased runtime, whereas $\tau = 50$ provided a 20% speedup with moderate accuracy loss. Balancing these trade-offs, we selected:

$$\alpha_0 = 0.2, \quad \gamma = 0.02, \quad \beta = 0.9, \quad \tau = 25.$$

The mini-batch size was set adaptively as $m = \min(200, \max(50, n/50))$. These hyperparameters, tuned exclusively on synthetic data, were held fixed for all real-data experiments.

**SMAVE sensitivity to neighborhood size and graph refresh period.** Figures 2–3 report a systematic sweep of both $k$ and $\tau$ hyperparameters across all six $(n, p) \in \{2000, 5000\} \times \{20, 50, 100\}$ configurations, averaged over the five link functions and both covariance structures (identity and AR(1)) with 10 replications each.

- For the neighbourhood size $k$, performance degrades sharply only when $k/k^* \lesssim 0.25$ (severely undersmoothed), while the error curve is nearly flat over $k/k^* \in [0.5, 3]$: halving or tripling the default $k^* = \lfloor n^{4/(d+4)} \rfloor$ changes $m^2$ by less than a factor of two in all settings. The flat region widens with $n$, consistent with the default entering the correct asymptotic regime.

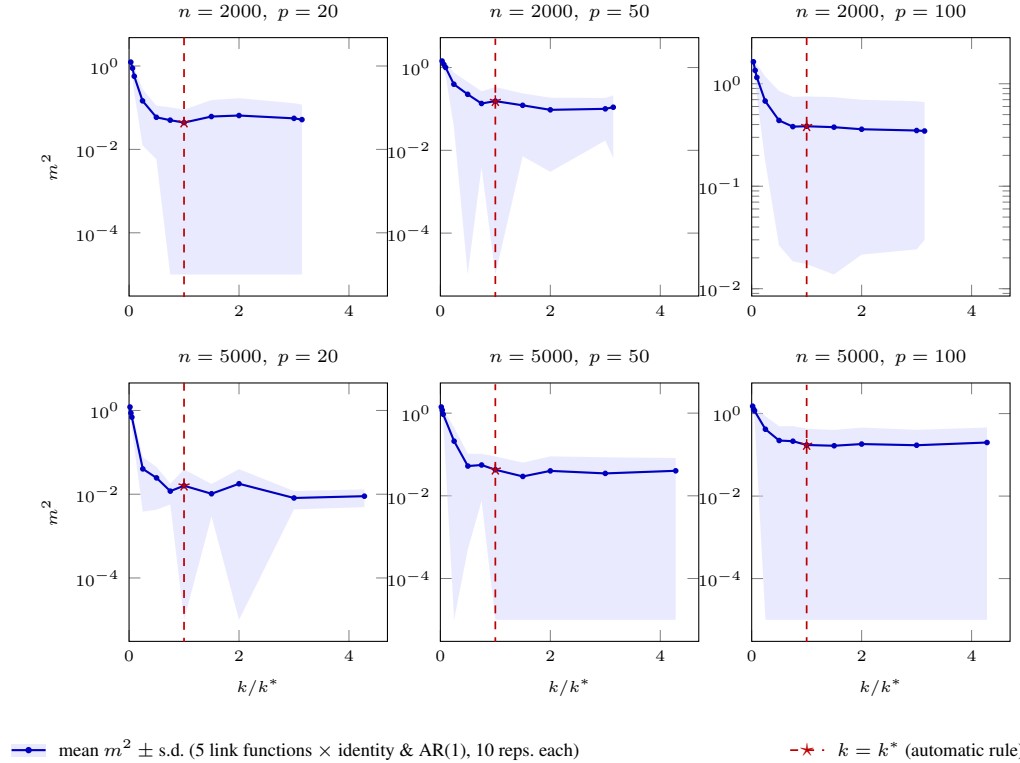

*Figure 2.* **Sensitivity of SMAVE to the neighbourhood size** $k$ ($d = 2$). Each panel shows mean $m^2 \pm$ one standard deviation, averaged over all 10 scenario combinations (five link functions $\times$ identity and AR(1) covariance) with 10 replications each. The $x$-axis is $k/k^*$, where $k^* = \lfloor n^{4/(d+4)} \rfloor$ (clipped to $[20, \lfloor n/3 \rfloor]$) is the theoretically motivated default. The red dashed line marks $k = k^*$; the red star shows the error of the automatic rule. Across all six $(n, p)$ configurations, $m^2$ deteriorates sharply for $k/k^* \lesssim 0.25$ (too few neighbours) and is nearly flat for $k/k^* \in [0.5, 3]$: halving or tripling $k^*$ changes accuracy only marginally, confirming that the rule is *not* a critical tuning decision. The flat region widens as $n$ increases, consistent with the $O(n^{4/(d+4)})$ default entering the correct asymptotic regime.

- For the refresh period $\tau$, accuracy is stable across $\tau \in [5, 40]$ in every panel, and only a frozen graph ($\tau = 100$) causes a clear increase in error. The default $\tau = 25$ lies well within the stable region.

Together, these results show that neither $k$ nor $\tau$ requires careful hand-tuning: the automatic rule for $k$ and the default $\tau$ are both robust choices.

## F. Extended Synthetic Experimental Results

This appendix reports the full five-method comparison underlying the ablation summary of Section 5, together with breakdowns by link function and covariance structure, robustness to random initialization, and behavior in the $p > n$ regime.

### F.1. Component ablation: full five-method comparison

Tables 7–8 report subspace recovery error and runtime for all five methods (OPG, OPG-$k$NN, RMAVE, RMAVE-$k$NN, SMAVE) on the full grid $n \in \{1000, 2000, 5000\}$, $p \in \{10, 20, 50, 100, 200\}$.

$k$-**NN localization does not improve RMAVE's accuracy.** RMAVE-$k$NN matches RMAVE almost exactly at low dimensions and is consistently slightly worse at higher dimensions. This is expected: the Gaussian kernel provides smooth, data-adaptive weights, which can be more informative for the local regression subproblems solved in RMAVE's coordinate descent than hard $k$-NN indicators.

*Table 7.* Squared subspace distance $m^2$ (mean $\pm$ s.e. over 10 scenarios $\times$ 10 replications). **Bold**: best; underline: second best.

| $n$ | $p$ | OPG | OPG-$k$NN | RMAVE | RMAVE-$k$NN | SMAVE |
|---|---|---|---|---|---|---|
| 1000 | 10 | $0.08_{\pm.01}$ | $0.04_{\pm.00}$ | $\mathbf{0.01}_{\pm\mathbf{.00}}$ | $0.01_{\pm.00}$ | $0.02_{\pm.00}$ |
| | 20 | $0.34_{\pm.02}$ | $0.24_{\pm.02}$ | $0.04_{\pm.01}$ | $\underline{0.03}_{\pm.00}$ | $0.06_{\pm.01}$ |
| | 50 | $1.90_{\pm.01}$ | $1.90_{\pm.01}$ | $0.54_{\pm.05}$ | $\underline{0.56}_{\pm.05}$ | $\mathbf{0.25}_{\pm\mathbf{.03}}$ |
| | 100 | $1.95_{\pm.00}$ | $1.21_{\pm.01}$ | $\underline{0.98}_{\pm.04}$ | $1.04_{\pm.04}$ | $\mathbf{0.59}_{\pm\mathbf{.04}}$ |
| | 200 | $1.96_{\pm.01}$ | $1.49_{\pm.01}$ | $\underline{1.48}_{\pm.03}$ | $1.54_{\pm.03}$ | $\mathbf{1.02}_{\pm\mathbf{.04}}$ |
| 2000 | 10 | $0.03_{\pm.00}$ | $0.02_{\pm.00}$ | $\mathbf{0.01}_{\pm\mathbf{.00}}$ | $0.01_{\pm.00}$ | $0.03_{\pm.01}$ |
| | 20 | $0.14_{\pm.01}$ | $0.11_{\pm.01}$ | $\mathbf{0.01}_{\pm\mathbf{.00}}$ | $0.01_{\pm.00}$ | $0.03_{\pm.00}$ |
| | 50 | $1.89_{\pm.01}$ | $1.87_{\pm.01}$ | $\underline{0.43}_{\pm.04}$ | $0.45_{\pm.05}$ | $\mathbf{0.11}_{\pm\mathbf{.02}}$ |
| | 100 | $1.95_{\pm.00}$ | $1.06_{\pm.01}$ | $\underline{0.73}_{\pm.04}$ | $0.77_{\pm.05}$ | $\mathbf{0.28}_{\pm\mathbf{.03}}$ |
| | 200 | $1.98_{\pm.00}$ | $1.37_{\pm.01}$ | $\underline{1.12}_{\pm.04}$ | $1.19_{\pm.04}$ | $\mathbf{0.59}_{\pm\mathbf{.04}}$ |
| 5000 | 10 | $0.01_{\pm.00}$ | $0.01_{\pm.00}$ | $\mathbf{0.00}_{\pm\mathbf{.00}}$ | $0.00_{\pm.00}$ | $0.02_{\pm.01}$ |
| | 20 | $0.05_{\pm.00}$ | $0.04_{\pm.00}$ | $\mathbf{0.00}_{\pm\mathbf{.00}}$ | $0.01_{\pm.00}$ | $0.01_{\pm.00}$ |
| | 50 | $1.89_{\pm.01}$ | $1.74_{\pm.03}$ | $0.33_{\pm.04}$ | $\underline{0.31}_{\pm.04}$ | $\mathbf{0.03}_{\pm\mathbf{.01}}$ |
| | 100 | $1.94_{\pm.00}$ | $0.82_{\pm.02}$ | $\underline{0.40}_{\pm.05}$ | $0.49_{\pm.05}$ | $\mathbf{0.13}_{\pm\mathbf{.03}}$ |
| | 200 | $1.98_{\pm.00}$ | $1.22_{\pm.01}$ | $\underline{0.69}_{\pm.05}$ | $0.74_{\pm.05}$ | $\mathbf{0.25}_{\pm\mathbf{.03}}$ |

*Table 8.* Runtime in seconds (mean over 10 scenarios $\times$ 10 replications). **Bold**: fastest; underline: second fastest.

| $n$ | $p$ | OPG | OPG-$k$NN | RMAVE | RMAVE-$k$NN | SMAVE |
|---|---|---|---|---|---|---|
| 1000 | 10 | 0.09 | **0.01** | 0.93 | 3.54 | 0.23 |
| | 20 | 0.14 | **0.02** | 1.52 | 3.94 | 0.19 |
| | 50 | 0.29 | **0.06** | 3.86 | 4.33 | 0.28 |
| | 100 | 0.68 | **0.13** | 5.83 | 5.98 | 0.49 |
| | 200 | 1.75 | **0.40** | 9.99 | 10.00 | 1.41 |
| 2000 | 10 | 0.32 | **0.03** | 3.27 | 14.56 | 0.26 |
| | 20 | 0.50 | **0.05** | 4.43 | 14.34 | 0.24 |
| | 50 | 1.04 | **0.11** | 12.94 | 15.67 | 0.35 |
| | 100 | 2.41 | **0.26** | 20.89 | 22.18 | 0.69 |
| | 200 | 6.25 | **0.79** | 34.89 | 35.20 | 1.78 |
| 5000 | 10 | 1.83 | **0.08** | 17.38 | 82.87 | 0.40 |
| | 20 | 2.91 | **0.15** | 20.19 | 80.26 | 0.50 |
| | 50 | 6.44 | **0.33** | 61.05 | 84.42 | 0.95 |
| | 100 | 14.16 | **0.72** | 98.01 | 115.93 | 2.02 |
| | 200 | 37.14 | **2.17** | 173.88 | 182.32 | 5.12 |

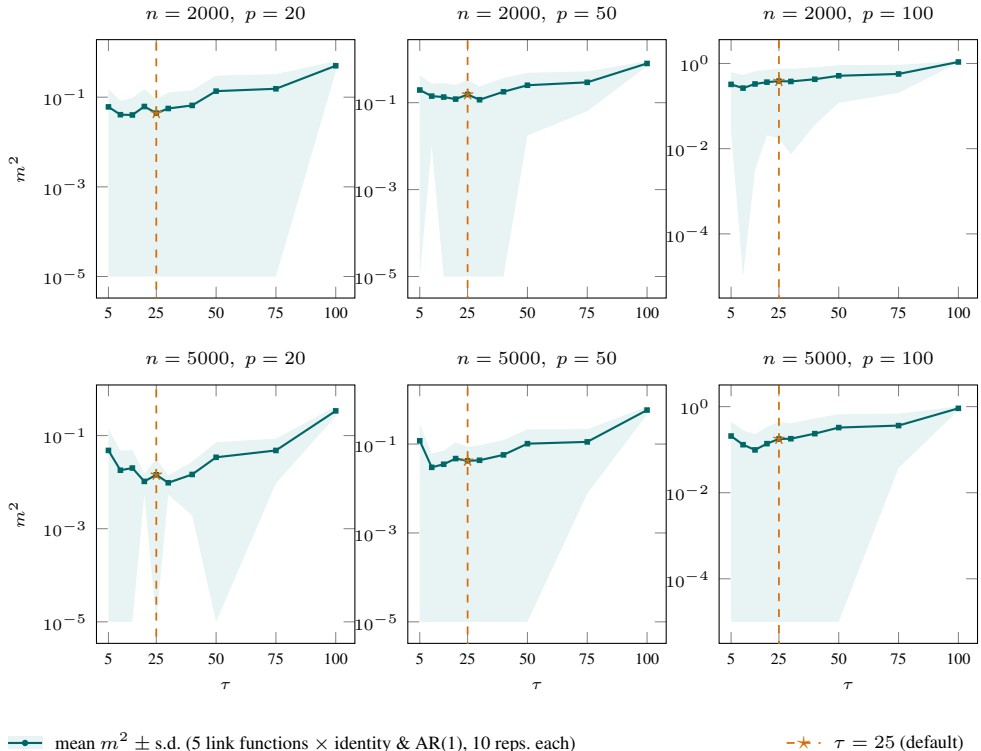

*Figure 3.* **Sensitivity of SMAVE to the $k$-NN refresh period $\tau$** ($d = 2$). $\tau$ is the number of gradient steps between successive rebuilds of the $k$-NN graph; $\tau = 25$ is the default. Each panel shows mean $m^2 \pm$ one standard deviation over all 10 scenario combinations (five link functions $\times$ identity and AR(1) covariance), with 10 replications each. The orange dashed line and star mark the default $\tau = 25$. Performance is stable across $\tau \in [5, 40]$ in all six $(n, p)$ configurations, while accuracy degrades noticeably only at $\tau = 100$ (a nearly frozen graph). This confirms that the exact refresh cadence is not a critical tuning parameter: refreshing every 20–50 steps yields indistinguishable accuracy, and the default $\tau = 25$ lies comfortably in the stable region.

**$k$-NN localization does not accelerate RMAVE.** Despite replacing dense kernel weights with sparse $k$-NN weights, RMAVE-$k$NN is actually *slower* than RMAVE. RMAVE's bottleneck lies in the column-wise coordinate-descent direction updates, which operate on the full $n \times n$ weight matrix at every outer iteration regardless of sparsity. Replacing the dense Gaussian kernel with $k$-NN weights requires, on top of the unchanged direction updates, computing all pairwise projected distances and selecting the $k$ smallest per anchor at every outer iteration. Both are quadratic in $n$ in our implementation. The sparse $k$-NN structure is not exploited by RMAVE's algorithmic structure, so the additional selection cost is incurred without any compensating speedup.

**Ambient-space $k$-NN is not enough.** Replacing OPG's kernel with ambient-space $k$-NN (OPG-$k$NN) slightly improves accuracy over OPG while reducing runtime but remains far from SMAVE performances. The conclusion is that $k$-NN *per se* is not what drives SMAVE's gains: the essential ingredient is $k$-NN localization *in the projected space $\mathbb{R}^d$*, combined with stochastic optimization and random initialization.

## F.2. Results by link function

We detail the five link functions $g : \mathbb{R}^2 \to \mathbb{R}$:

$$g_1(\mathbf{z}) = z_1(z_1 + z_2 + 1) \qquad \text{(polynomial)},$$
$$g_2(\mathbf{z}) = \sin(z_1) + \tfrac{1}{2}\cos(2z_2) \qquad \text{(sinusoidal)},$$
$$g_3(\mathbf{z}) = \exp(-z_1^2) + \tfrac{1}{2}z_2 \qquad \text{(exponential)},$$
$$g_4(\mathbf{z}) = z_1 z_2 + \sin(z_1) + \exp(-z_2^2/2) \qquad \text{(interaction)},$$
$$g_5(\mathbf{z}) = z_1/(0.5 + (z_2 + 1)^2) \qquad \text{(rational)}.$$

These span polynomial, periodic, localized, and rational nonlinearities, each paired with two covariance structures (identity and AR(1) with $\Sigma_{ij} = 0.5^{|i-j|}$), yielding 10 scenarios.

Table 9 disaggregates performance by link. SMAVE achieves the lowest error across all five, with particularly strong gains on symmetric nonlinearities (sinusoidal, exponential) where gradient magnitudes vary substantially across the input domain. OPG shows little variation across links, confirming that its limiting factor is ambient-space localization rather than link complexity.

*Table 9.* Squared subspace distance $m^2$ by link function (mean $\pm$ s.e. over $n$, $p$, covariance, replications). **Bold**: best; underline: second best.

| Link | OPG | RMAVE | SMAVE |
|------|-----|-------|-------|
| Polynomial | $0.75_{\pm.06}$ | $0.19_{\pm.03}$ | $\mathbf{0.07}_{\pm.01}$ |
| Sinusoidal | $0.70_{\pm.07}$ | $\underline{0.27}_{\pm.03}$ | $\mathbf{0.15}_{\pm.02}$ |
| Exponential | $0.71_{\pm.06}$ | $\underline{0.22}_{\pm.03}$ | $\mathbf{0.09}_{\pm.01}$ |
| Interaction | $0.67_{\pm.07}$ | $\underline{0.02}_{\pm.01}$ | $\mathbf{0.02}_{\pm.00}$ |
| Rational | $0.71_{\pm.06}$ | $\underline{0.08}_{\pm.02}$ | $\mathbf{0.03}_{\pm.00}$ |

## F.3. Results by covariance structure

All methods exhibit similar relative rankings under identity and AR(1) covariance. SMAVE achieves comparable error under identity ($0.08_{\pm.01}$) and AR(1) ($0.06_{\pm.01}$), indicating that $k$-NN localization is robust to correlation structure without requiring explicit covariance adaptation.

*Table 10.* Squared subspace distance $m^2$ by covariance structure (mean $\pm$ s.e. over link functions, $(n,p)$, replications). **Bold**: best; underline: second best.

| Covariance | OPG | RMAVE | SMAVE |
|------------|-----|-------|-------|
| Identity | $0.70_{\pm.04}$ | $0.16_{\pm.02}$ | $\mathbf{0.08}_{\pm.01}$ |
| AR(1) | $0.72_{\pm.04}$ | $\underline{0.15}_{\pm.02}$ | $\mathbf{0.06}_{\pm.01}$ |

## F.4. Robustness to random initialization

A central design choice of SMAVE is to drop the OPG warm start used by RMAVE in favor of a uniform draw on $\mathrm{St}(p, d)$. To verify that this is reliable, we ran SMAVE with 100 independent random Stiefel initializations on each of the 10 scenarios at $(n, p) = (5000, 100)$ (1,000 runs total), and compared against the original 10-replication experiment at the same configuration.

*Table 11.* SMAVE robustness to random initialization at $(n, p) = (5000, 100)$.

| | 100 inits | 10 inits |
|---|-----------|----------|
| Mean $m^2_{\pm\text{s.e.}}$ | $0.11_{\pm 0.01}$ | $0.14_{\pm 0.03}$ |

The two estimates agree, confirming that 10 initializations already give a reliable picture of SMAVE's performance. Variability is concentrated in two identity-covariance scenarios (sinusoidal and exponential) where gradient outer products

exhibit near-cancellation; across the remaining eight scenarios, standard deviation across initializations is at most 0.02. Together with the convergence curves (Appendix F.6), this supports the picture that the SMAVE objective (6) has a benign enough optimization landscape for random initialization to be sufficient, and that the stochastic updates do useful exploration rather than merely adding noise.

## F.5. Behavior in the $p > n$ regime

*Table 12.* Mean squared subspace distance $m^2$ in the $p > n$ regime. All methods return near-random estimates; we report SMAVE's robustness rather than its outperformance. $^\dagger$OPG-$k$NN crashes at $n = 50$ because $k > n$.

| $n$ | $p$ | OPG | OPG-$k$NN | RMAVE | RMAVE-$k$NN | SMAVE |
|---|---|---|---|---|---|---|
| 50 | 100 | 1.952 | —$^\dagger$ | 1.870 | 1.898 | 1.820 |
| 50 | 200 | 1.975 | —$^\dagger$ | 1.919 | 1.906 | 1.903 |
| 100 | 200 | 1.974 | 1.843 | 1.902 | 1.937 | 1.857 |

We additionally ran all five methods on $n \in \{50, 100\}$, $p \in \{100, 200\}$, $d = 2$ (10 scenarios $\times$ 10 replications). Without sparsity assumptions, the $p > n$ regime is information-theoretically hard for the methods considered, and all return near-random estimates ($m^2 \approx 1.8$–$2.0$ out of $d = 2$). We report this configuration to verify that SMAVE *degrades gracefully* rather than failing: OPG-$k$NN crashes at $n = 50$ because $k > n$, while SMAVE's clipping rule $k \in [20, \lfloor n/3 \rfloor]$ and projected-space localization in $\mathbb{R}^d$ handle the regime without modification.

## F.6. Convergence curves

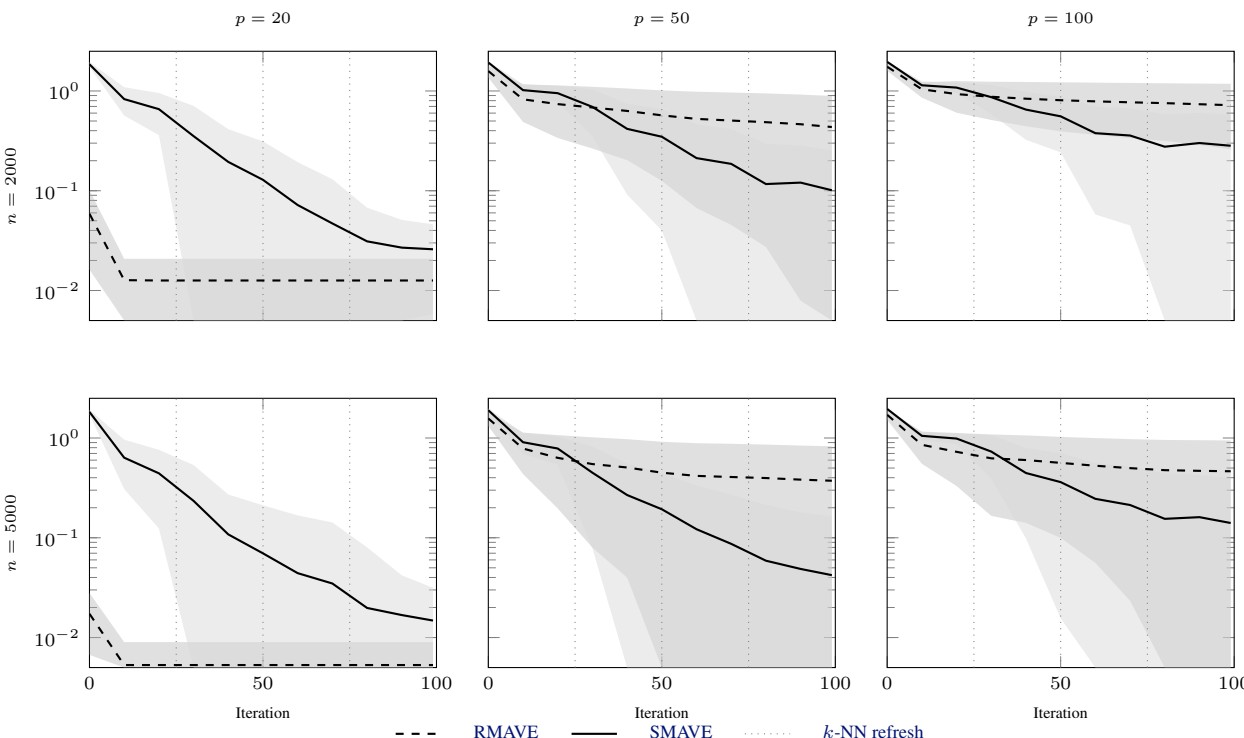

*Figure 4.* Convergence of subspace error $m^2$ for $n \in \{2000, 5000\}$ (rows) and $p \in \{20, 50, 100\}$ (columns). Each curve shows the mean $\pm 1$ std aggregated over 5 link functions, 2 covariance structures, and 10 replications (100 runs total). Dotted vertical lines indicate $k$-NN refresh iterations for SMAVE. At low dimension ($p = 20$), RMAVE converges within a few iterations; at higher dimensions ($p \geq 50$), SMAVE achieves 2–9$\times$ lower final error.

# G. Real-data Experiments: Test MSE by Candidate Dimension

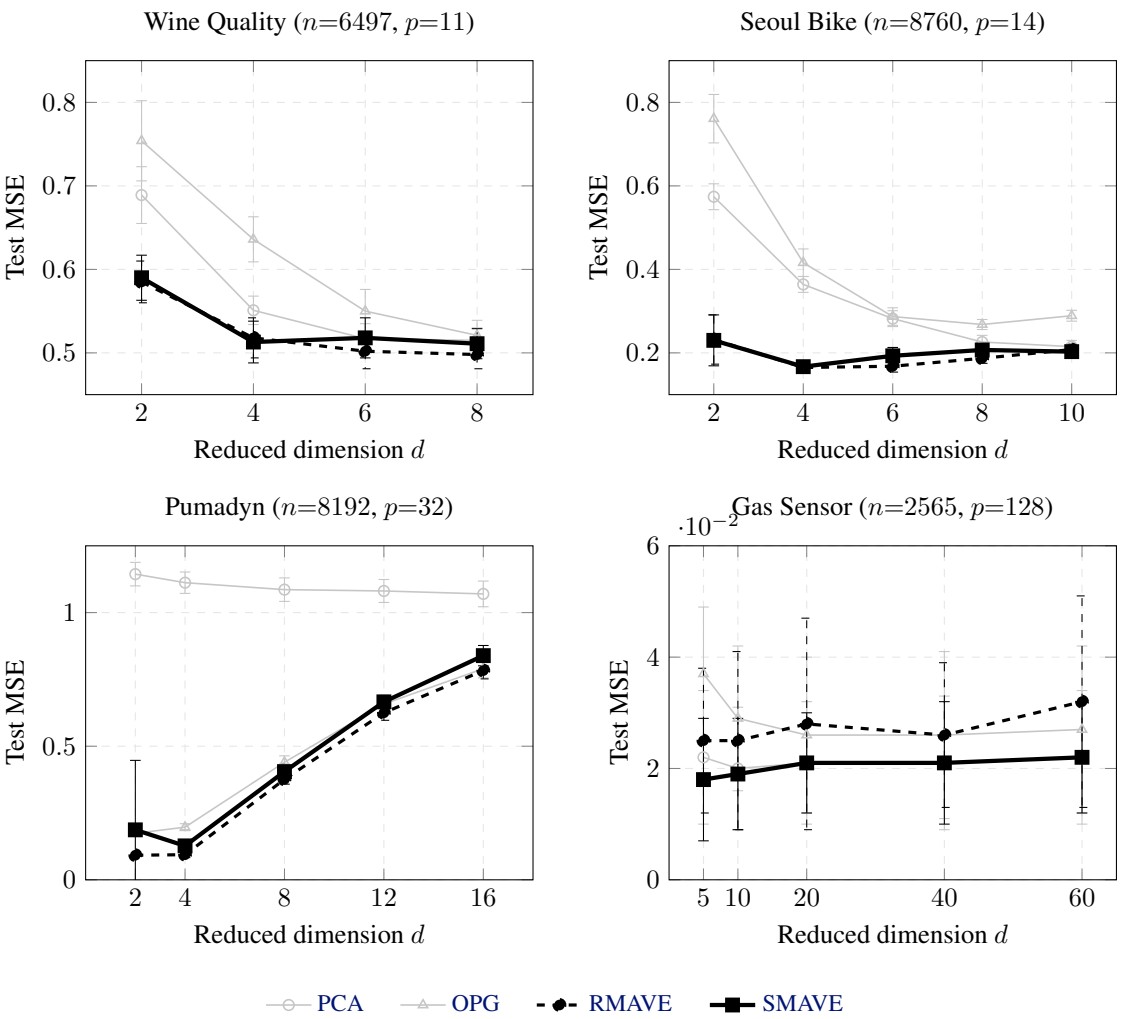

*Figure 5.* Test MSE vs. reduced dimension $d$ across four datasets. Baselines (PCA, OPG) in gray; iterative methods (RMAVE, SMAVE) in black.

