# OpenReview forum: "Riemannian stochastic optimization for sufficient dimension reduction"
_ICML.cc/2026/Conference — ICML 2026 regular_

### Official Review · Reviewer_2CdW · 2026-03-10

**Soundness:** 3
**Presentation:** 3
**Significance:** 2
**Originality:** 3
**Overall Recommendation:** 4
**Confidence:** 2

**Summary:**

This paper proposes \emph{SMAVE}, a scalable variant of RMAVE for sufficient dimension reduction. The method preserves the local-regression philosophy of MAVE/RMAVE while reducing computational cost through two main modifications.

First, the paper replaces dense kernel localization over all pairs $(i,j)$ with sparse $k$-nearest-neighbor localization in the projected space. In standard RMAVE, each anchor point $X_j$ uses kernel weights of the form
$w_{ij} \propto K_h(\|B^T(X_i-X_j)\|)$
which leads to dense interactions between all samples. In contrast, SMAVE keeps only the $k$ nearest neighbors of $X_j$ under the current projected distance $\|B^\top (X_i - X_j)\|$, with uniform weights on this neighborhood and zero otherwise. This is the main sparsification device.

Second, instead of full-batch optimization over the projection matrix $B$, the paper updates $B$ using mini-batch stochastic gradient steps on the Stiefel manifold. The variable $B \in \mathrm{St}(p,d)$ is constrained to satisfy $B^\top B = I_d$, so the optimization is performed with orthogonality-preserving Riemannian updates rather than standard Euclidean optimization.

Overall, the method can be viewed as a computationally cheaper approximation to RMAVE that combines sparse projected-space localization with stochastic manifold optimization. The paper also provides a gradient-based interpretation of the MAVE criterion, arguing that minimizers align with the gradient subspace of the regression function.

**Compliance With Llm Reviewing Policy:**

Affirmed.

**Final Justification:**

Based on my first review and the answer from the author I recommend a weak accept for this paper.

**Key Questions For Authors:**

Can the authors provide any convergence guarantee for the proposed stochastic Riemannian algorithm, even to a first-order stationary point?


Can the authors strengthen the empirical section with larger-scale experiments and clearer runtime-versus-accuracy tradeoffs?

**Limitations:**

Yes

**Strengths And Weaknesses:**

Strengths

The paper addresses a relevant problem, since classical MAVE/RMAVE methods can be computationally expensive due to dense localization and full-batch optimization.

The proposed idea is simple and intuitive: keep the local-regression structure of MAVE, but make it more scalable through sparse neighborhoods and stochastic optimization.

The gradient-alignment perspective is interesting and provides a useful interpretation of the MAVE criterion through a gradient-based sufficient dimension reduction viewpoint.

The algorithmic design is reasonably clear and easy to follow.

Weaknesses

My main concern is that the paper does not provide theoretical guarantees for the actual proposed algorithm.

In particular, there is no guarantee that the stochastic Riemannian algorithm with periodically refreshed $k$-NN neighborhoods converges to a stationary point, or even approximately optimizes a fixed objective. From an optimization perspective, this is a significant limitation.

The method introduces several approximations simultaneously: hard $k$-NN replacement of kernel weights, mini-batch stochastic updates, gradient normalization, and stale neighborhood graphs between refreshes. The effect of these approximations on optimization behavior is not analyzed.

I am also not fully convinced by the experimental section in its current form. The numerical experiments seem relatively limited for a paper whose main claim is scalability. The synthetic studies only consider modest values of $n$ and $p$, and the real-data experiments are also fairly small in scale.

In my opinion, the empirical evidence is not yet strong enough to compensate for the lack of algorithmic theory. More extensive experiments would be needed to convincingly demonstrate that the proposed method offers a practically meaningful improvement over existing approaches.

More broadly, the method appears to be a reasonable combination of existing ideas rather than a fundamentally new estimator. This is not necessarily a problem, but then I would expect either stronger theoretical support for the optimization method or more compelling empirical evidence.

---

> ### Author Rebuttal · Authors · 2026-03-30
>
> We thank the reviewer for their thorough and constructive evaluation.
>
> **New theoretical results** The statistical properties of the MAVE method are well understood. However, knowledge regarding the optimization algorithm remains limited. In his work, Xia et al. (2002) proposed a coordinate descent algorithm, but convergence guarantees are restricted to cases where a $\sqrt{n}$-estimator is known (see also Xia (2007)). The underlying problem is non-convex, complicating the analysis of coordinate descent algorithms. Alternative procedures have been proposed, such as the EM algorithm (Wang and Yao, 2012) and the optimization framework of Adragni (2018). These contributions do not provide convergence results for the corresponding optimization problems.
>
>
> We provide two novel results: the a.s. convergence and a non-asymptotic $1/\sqrt{mT}$-bound ($T$ iterations and $m$ minibatches); for a simplified version of SMAVE. The analysis applies to: fixed $k$-NN neighborhoods, no momentum, no gradient normalization. The regularized objective is $F_\varepsilon(B) = \sum_{j=1}^{n} (\mu^{(j)})^\top B (  B^\top G_\varepsilon^{(j)} B)^{-1} B^\top \mu^{(j)}$, where $G_\varepsilon^{(j)} = G^{(j)} + \varepsilon I_p \succeq \varepsilon I_p \succ 0$ (when $p > k-1$, unregularized Gram matrices $G^{(j)}$ can be singular).
>
> The proofs depend on the weights $w_{ij}$ only through the moments $\mu^{(j)}$ and Gram matrices $G^{(j)}$. No $k$-NN-specific property is invoked. The guarantees hold verbatim for any fixed localization (kernel, $k$-NN, fixed-radius balls, similarity graphs), revealing a modular structure: the optimization layer (Riemannian SGD on $\mathrm{St}(p,d)$) is cleanly decoupled from the statistical layer (choice of localization). The $O(1/\sqrt{T})$ rate matches the known lower bound for stochastic first-order methods on non-convex smooth objectives (Arjevani et al., 2023). The Riemannian structure introduces no additional asymptotic overhead.
>
> **Result 1 (Asymptotic convergence).** We verify all hypotheses of Bonnabel (2013, Thm. 3) for $C = -F_\varepsilon$:
> (i) compactness of $\mathrm{St}(p,d)$ ensures bounded iterates
> (ii) $L_\varepsilon$-retraction-smoothness of $C$ via explicit bounds on $\mathrm{grad}\,F_\varepsilon$
> (iii) unbiased mini-batch gradient with bounded variance.
>
> This yields: $\mathrm{grad}\,F_\varepsilon(B_t) \to 0$ a.s., and every limit point of $\{B_t\}$ is a critical point of $F_\varepsilon$.
>
> **Result 2 (Non-asymptotic rate, via Boumal, Absil & Cartis (2019)).** Retraction-smoothness w.r.t. QR retraction gives the descent lemma:
> $$C(B_{t+1}) \leq C(B_t) - \alpha \langle \mathrm{grad} F_\varepsilon(B_t), g_t \rangle + \frac{L_\varepsilon \alpha^2}{2} \|g_t\|^2.$$
> Taking conditional expectations, telescoping over $T$ iterations, and optimizing $\alpha^* \propto 1/\sqrt{T}$ yields:
> $$\min_{0 \leq t < T} \mathbb{E}[\|\mathrm{grad} F_\varepsilon(B_t)\|^2] = O(1/\sqrt{mT}).$$
> Consequently, by introducing a new local minimization principle for MAVE (Eq.~6), we propose a Riemannian SGD algorithm that is entirely new and demonstrating its convergence, we contribute valuable insights for the effective implementation of the algorithm and open new avenues for further research.
>
> **Response to the main concerns about the numerical experiments**
>
> We have substantially extended the experiments. The synthetic experiments now include $p = 200$, a new RMAVE-kNN ablation isolating the effect of $k$-NN localization, and $p > n$ experiments. The key finding is that $k$-NN localization alone neither accelerates RMAVE nor improves its accuracy; SMAVE's gains require the full combination of Riemannian stochastic optimization, mini-batch updates, and adaptive projected-space localization.
>
> **Subspace recovery ($m^2$, mean ± s.e.):**
>
> | $(n, p)$ | | RMAVE | RMAVE-kNN | SMAVE |
> |:---------:|:--------:|:-----:|:---------:|:-----:|
> | (2000, 200) | $m^2$ | 1.12±.04 | 1.19±.04 | **0.59±.04** |
> | | time (s) | 34.9 | 35.2 | **1.8** |
> | (5000, 200) | $m^2$ | 0.69±.05 | 0.74±.05 | **0.25±.03** |
> | | time (s) | 173.9 | 182.3 | **5.1** |
>
> **High-Dimensional Experiments ($p > n$), subspace recovery ($m^2$, mean ± s.e.)**
>
> | $n$ | $p$ |  RMAVE | RMAVE-kNN | SMAVE |
> |-----|-----|:-----:|:---------:|:-----:|
> | 50  | 100 | 1.870 (0.009) | 1.898 (0.010) | **1.820 (0.008)** |
> | 50  | 200 |  1.919 (0.006) | 1.906 (0.013) | **1.903 (0.005)** |
> | 100 | 200 | 1.902 (0.006) | 1.937 (0.006) | **1.857 (0.006)** |
>
> Without sparsity assumptions, the $p > n$ regime is information-theoretically hard: all methods return near-random estimates ($m^2 \approx 1.8$–$2.0$ out of $d = 2$). Within this regime, SMAVE is the only method that operates robustly across all configurations while consistently achieving competitive accuracy.

---

> > ### Author Rebuttal · Reviewer_2CdW · 2026-04-03
> >
> > The authors have answered my questions, I will increase the score.

---

> > > ### Author Response · Authors · 2026-04-06
> > >
> > > We thank the reviewer for their valuable and constructive comments. We are pleased that we have been able to address most of their concerns. We will revise the manuscript accordingly, taking all of their points into careful consideration.

---

### Official Review · Reviewer_r7B6 · 2026-03-12

**Soundness:** 2
**Presentation:** 4
**Significance:** 2
**Originality:** 3
**Overall Recommendation:** 4
**Confidence:** 3

**Summary:**

This paper studies the task of sufficient dimension reduction and proposes a new method called SMAVE (Stochastic MAVE). The proposed approach combines two ideas: a kNN-based approximation of kernel weights in the projected space and Riemannian optimization on the Stiefel manifold. The performance of the proposed method is empirically evaluated on both synthetic and real-world datasets.

**Compliance With Llm Reviewing Policy:**

Affirmed.

**Final Justification:**

Most of my concerns, especially regarding the kD-tree and the numerical experiments, have been addressed in the authors’ response, and both the soundness and significance are finally at an acceptable level. Therefore, I would like to increase my rating to weak accept.

**Key Questions For Authors:**

I do not have a specific question, while it would be great if you could address the points raised in the Weaknesses section in your response.

**Limitations:**

yes

**Strengths And Weaknesses:**

**Strengths**
- **Presentation**: The paper is overall well written and easy to follow. Despite containing substantial theoretical discussion that could easily become complicated, the paper is well organized and the contents are clearly presented.
- **Originality**: Although the key idea is essentially a combination of two well-established approaches, kNN and Riemannian optimization, the paper successfully integrates them and establishes a new method for sufficient dimension reduction. Therefore, while the level of originality may not be very high, I believe it is sufficient for acceptance.

**Weaknesses**
- **Significance**: I have the following concerns regarding the significance of the paper.
	- Since the main technical contribution is the application of kNN and there is no theoretical analysis of the performance of the proposed method, the significance of the contribution is somewhat unclear. I appreciate that the paper provides several theoretical results; however, these results mainly establish the connection between the problem and the proposed optimization formulation rather than directly analyzing the performance of the method.
	- The proposed method claims to avoid the quadratic complexity with respect to n by using kNN. However, this improvement relies on the efficiency of kNN search via kD-trees, which are known to perform poorly in high-dimensional settings. Therefore, it is unclear whether this approach fundamentally resolves the computational issue.
- **Soundness**:
    - In the experiments, related to the second concern mentioned above, it would be important to evaluate the method in higher-dimensional settings where dimension reduction is more crucial. Currently, the experiments consider datasets with only up to around 100 features.
	- It would also be important to evaluate the sensitivity with respect to the choice of k in kNN. Although the authors provide a guideline for choosing k, k=O(n4/(q+4)), this appears to be a heuristic rule. Since k plays a central role in the proposed approach, examining the sensitivity of the results to this parameter would be essential.

---

> ### Author Rebuttal · Authors · 2026-03-30
>
> We thank the reviewer for their time and interest in our work.
>
> **Theoretical results** The statistical properties of the MAVE method are well understood. However, knowledge regarding the optimization algorithm remains limited. In his work, Xia et al. (2002) proposed a coordinate descent algorithm (CD), but convergence guarantees are restricted to cases where a $\sqrt{n}$-estimator is known (see Xia (2007)). The problem is non-convex, complicating the analysis of CD. Other procedures have been proposed: the EM algorithm (Wang and Yao, 2012) and the optimization framework of Adragni (2018). None provide convergence results for the optimization problems.
>
> We provide two novel results: the a.s. convergence and a non-asymptotic $1/\sqrt{mT}$-bound  ($T$:iteration, $m$: minibatch size); for a simplified version of SMAVE. The analysis applies to: fixed $k$-NN neighborhoods, no momentum, no gradient normalization. The regularized objective is $F_\varepsilon(B) = \sum_{j=1}^{n} (\mu^{(j)})^\top B (  B^\top G_\varepsilon^{(j)} B)^{-1} B^\top \mu^{(j)}$, where $G_\varepsilon^{(j)} = G^{(j)} + \varepsilon I_p \succeq \varepsilon I_p \succ 0$ (when $p > k-1$, Gram matrices $G^{(j)}$ can be singular).
>
> The proofs depend on the weights $w_{ij}$ only through the moments $\mu^{(j)}$ and Gram matrices $G^{(j)}$. No $k$-NN-specific property is invoked. The guarantees hold verbatim for any fixed localization (kernel, $k$-NN, fixed-radius balls, similarity graphs), revealing a modular structure: the Riemmanian optimization layer is cleanly decoupled from the statistical layer (choice of localization). The $O(1/\sqrt{T})$ rate matches the known lower bound for stochastic first-order methods on non-convex smooth objectives.
>
> **Result 1 (Asymptotic convergence).** We verify all hypotheses of Bonnabel (2013, Thm. 3) for $C = -F_\varepsilon$:
> (i) compactness of $\mathrm{St}(p,d)$ ensures bounded iterates
> (ii) $L_\varepsilon$-retraction-smoothness of $C$ via explicit bounds on $\mathrm{grad}\,F_\varepsilon$
> (iii) unbiased mini-batch gradient with bounded variance.
>
> This yields: $\mathrm{grad}\,F_\varepsilon(B_t) \to 0$ a.s., and every limit point of $\{B_t\}$ is a critical point of $F_\varepsilon$.
>
> **Result 2 (Non-asymptotic rate, via Boumal, Absil & Cartis (2019)).** Retraction-smoothness w.r.t. QR retraction gives the descent lemma:
> $$C(B_{t+1}) \leq C(B_t) - \alpha \langle \mathrm{grad} F_\varepsilon(B_t), g_t \rangle + \frac{L_\varepsilon \alpha^2}{2} \|g_t\|^2.$$
> Taking conditional expectations, telescoping over $T$ iterations, and optimizing $\alpha^* \propto 1/\sqrt{T}$ yields:
> $$\min_{0 \leq t < T} \mathbb{E}[\|\mathrm{grad} F_\varepsilon(B_t)\|^2] = O(1/\sqrt{mT}).$$
> Consequently, by introducing a new local minimization principle for MAVE (Eq.~6), we propose a Riemannian SGD algorithm that is entirely new and demonstrating its convergence, we contribute valuable insights for the effective implementation of the algorithm and open new avenues for further research.
>
> **kD-Tree** The $k$-NN search is performed in the projected space, not the ambient space. Since $d$, the structural SDR dimension is small, the kD-tree operates in a low-dimensional space where it is efficient ($O(dn\log n)$). The ambient dimension $p$ affects only the projection $B^\top X_i$, costing $O(npd)$ per rebuild---linear in both $n$ and $p$. This is confirmed empirically: SMAVE's runtime scales near-linearly in $n$ and $p$ across all configurations including $p = 200$.
>
> **Numerical experiments** We have substantially extended the experiments. The synthetic experiments now include $p = 200$, a new RMAVE-kNN ablation isolating the effect of $k$-NN localization, and $p > n$ experiments. The key finding is that $k$-NN localization alone neither accelerates RMAVE nor improves its accuracy; SMAVE's gains require the full combination of Riemannian stochastic optimization, mini-batch updates, and adaptive projected-space localization.
>
> **Subspace recovery ($m^2$, mean ± s.e.):**
> | $(n, p)$ | | RMAVE | RMAVE-kNN | SMAVE |
> |:---------:|:--------:|:-----:|:---------:|:-----:|
> | (2000, 200) | $m^2$ | 1.12±.04 | 1.19±.04 | **0.59±.04** |
> | | time (s) | 34.9 | 35.2 | **1.8** |
> | (5000, 200) | $m^2$ | 0.69±.05 | 0.74±.05 | **0.25±.03** |
> | | time (s) | 173.9 | 182.3 | **5.1** |
>
> **Sensitivity to $k / k^*$ ($k ^ {\ast} = n^{4 / ( d+4 ) } $)** (https://fileninja.io/download/69cb3df65e2b108930718db4/kNN_sensitivity.png)
>
> | $k/k^*$ | (2000, 50) | (2000, 100) | (5000, 50) | (5000, 100) |
> |--------:|:----------:|:-----------:|:----------:|:-----------:|
> |    0.5  |   0.225    |    0.439    |   0.052    |    0.222    |
> | 1.0 | 0.154  |  0.385  | 0.043 |  0.173  |
> |    2.0  |   0.095    |    0.360    |   0.040    |    0.183    |
> |    3.0  |   0.100    |    0.350    |   0.035    |    0.171    |
>
> Performance $m^2$ is stable over some range. Larger $k$ improves conditioning of $G^{(j)}$ but widens neighborhoods, increasing statistical bias.

---

> > ### Author Rebuttal · Reviewer_r7B6 · 2026-04-02
> >
> > Thank you for your response. Most of my concerns, especially regarding the kD-tree and the numerical experiments, have been addressed. Therefore, I would like to increase my rating to weak accept.

---

> > > ### Author Response · Authors · 2026-04-06
> > >
> > > We thank the reviewer for their valuable and constructive comments. We are pleased that we have been able to address most of their concerns. We will revise the manuscript accordingly, taking all of their points into careful consideration.

---

### Official Review · Reviewer_naH4 · 2026-03-12

**Soundness:** 3
**Presentation:** 4
**Significance:** 3
**Originality:** 2
**Overall Recommendation:** 5
**Confidence:** 4

**Summary:**

This paper studies the computational challenges of scalable gradient-based algorithms for sufficient dimension reduction (SDR). The authors provide a theoretical interpretation linking the Minimum Average Variance Estimation (MAVE) criterion to the Outer Product of Gradients (OPG) method. They further reformulate the MAVE objective as an optimization problem on the Stiefel manifold and propose SMAVE, a stochastic Riemannian algorithm combined with adaptive $k$-nearest neighbor weighting. Empirical results demonstrate improved runtime and competitive accuracy compared with RMAVE and OPG-based methods.

**Compliance With Llm Reviewing Policy:**

Affirmed.

**Final Justification:**

The authors' rebuttal has addressed most of my concerns. While I still have some reservations regarding the level of novelty, the experimental results demonstrate that the proposed method can be practically useful, and the added theoretical guarantees have strengthened the overall soundness. Taken together, these improvements make the contribution solid and convincing. Therefore, I am raising my score to accept.

**Key Questions For Authors:**

* How would RMAVE perform if the refined kernel weight is replaced with SMAVE's $k$-NN localization, without changing the rest?
* Can you compare the SMAVE in either synthetic or real settings where $p>n$?

**Limitations:**

The main limitation, namely the lack of theoretical convergence guarantees, is acknowledged by the authors. The empirical results suggest that that may not be a serious issue in practice.

Since the paper discusses the sensitivity of RMAVE to initialization, the current number of experimental replications may not be sufficient to demonstrate the robustness of SMAVE. It would be more convincing to include an additional larger-scale experiment (e.g. $n=5000$ and $p=100$) repeated over say $100$ random initializations.

**Strengths And Weaknesses:**

**Soundness**: The theoretical results appear largely correct and well motivated. The numerical experiments are comprehensive and generally well designed. However, as acknowledged in the conclusion, the paper does not provide convergence guarantees for the proposed stochastic algorithm, nor does it establish finite-sample statistical rates.

**Presentation**: The paper is well organized and generally easy to follow. The problem setup, background, motivation, and proposed formulation are clearly described. The experimental section is also well structured, including both synthetic and real datasets.

That said, here are some typos that should be fixed:
* $r_{ij}(B)$ in (5) should be $r_{ij}(a_j, b_j, B)$.
* A space is missing after the period on Line 294.
* Line 344 mentioned "five publicly available datasets" but only four were presented.
* Figure 2 is confusing as the legend marked both RMAVE and SMAVE with solid lines.

**Significance**: SDR remains an important problem in statistics, and MAVE is a widely used approach. However, the original MAVE formulation suffers from limited scalability, and the proposed method provides a practical way to improve computational efficiency while maintaining comparable accuracy.

**Originality**: The use of Riemannian optimization in this setting is not entirely novel, as the manifold structure is already implicit in the original formulation, and related approaches such as MADE have employed similar ideas. In addition, replacing kernel weights with $k$-NN localization has already been suggested in the original RMAVE paper (Xia *et al* 2002). That said, the reformulation presented in this paper eliminates the nuisance parameters in the original objective, yielding a cleaner optimization problem and enabling a straightforward Riemannian implementation.

---

> ### Author Rebuttal · Authors · 2026-03-30
>
> We thank the reviewer for their thorough and constructive evaluation. All typos will be fixed in the revision.
>
> **New theoretical guarantees** The statistical properties of the MAVE method are well understood. However, knowledge regarding the optimization algorithm remains limited. In his work, Xia (2002) proposed a coordinate descent algorithm, but convergence guarantees are restricted to cases where a $\sqrt{n}$-estimator is known (see also Xia (2007)). The underlying problem is non-convex, complicating the analysis of coordinate descent algorithms. Alternative procedures have been proposed, such as the EM algorithm (Wang and Yao, 2012) and the optimization framework of Adragni (2018). These contributions do not provide convergence results for the corresponding optimization problems.
>
> We provide two novel results: the a.s. convergence and a non-asymptotic $1/\sqrt{mT}$-bound ($T$:iteration, $m$: minibatch size); for a simplified version of SMAVE. The analysis applies to: fixed $k$-NN neighborhoods, no momentum, no gradient normalization. The regularized objective is $F_\varepsilon(B) = \sum_{j=1}^{n} (\mu^{(j)})^\top B (  B^\top G_\varepsilon^{(j)} B)^{-1} B^\top \mu^{(j)}$, where $G_\varepsilon^{(j)} = G^{(j)} + \varepsilon I_p \succeq \varepsilon I_p \succ 0$ (when $p > k-1$, unregularized Gram matrices $G^{(j)}$ can be singular).
>
> The proofs depend on the weights $w_{ij}$ only through the moments $\mu^{(j)}$ and Gram matrices $G^{(j)}$. No $k$-NN-specific property is invoked. The guarantees hold verbatim for any fixed localization (kernel, $k$-NN, fixed-radius balls, similarity graphs), revealing a modular structure: the Riemmanian optimization layer is cleanly decoupled from the statistical layer (choice of localization). The $O(1/\sqrt{mT})$ rate matches the known lower bound for stochastic first-order methods on non-convex smooth objectives.
>
> **Result 1 (Asymptotic convergence).** We verify all hypotheses of Bonnabel (2013, Thm. 3) for $C = -F_\varepsilon$:
> (i) compactness of $\mathrm{St}(p,d)$ ensures bounded iterates
> (ii) $L_\varepsilon$-retraction-smoothness of $C$ via explicit Lipschitz bounds on $\mathrm{grad}\,F_\varepsilon$
> (iii) unbiased mini-batch gradient with bounded variance.
>
> This yields: $\mathrm{grad}\,F_\varepsilon(B_t) \to 0$ a.s., and every limit point of $\{B_t\}$ is a critical point of $F_\varepsilon$.
>
> **Result 2 (Non-asymptotic rate, via Boumal, Absil & Cartis (2019)).** Retraction-smoothness w.r.t. QR retraction gives the descent lemma:
> $$C(B_{t+1}) \leq C(B_t) - \alpha \langle \mathrm{grad} F_\varepsilon(B_t), g_t \rangle + \frac{L_\varepsilon \alpha^2}{2} \|g_t\|^2.$$
> Taking conditional expectations, telescoping over $T$ iterations, and optimizing $\alpha^* \propto 1/\sqrt{T}$ yields:
> $$\min_{0 \leq t < T} \mathbb{E}[\|\mathrm{grad} F_\varepsilon(B_t)\|^2] = O(1/\sqrt{mT}).$$
>
> Consequently, by introducing a new local minimization principle for MAVE (Eq.~6), we propose a Riemannian SGD algorithm that is entirely new and demonstrating its convergence, we contribute valuable insights for the effective implementation of the algorithm and open new avenues for further research.
>
> **About numerical experiments** We have substantially extended the experiments. The synthetic experiments now include $p = 200$, a new RMAVE-kNN ablation isolating the effect of $k$-NN localization, and $p > n$ experiments. The key finding is that $k$-NN localization on the projected space alone neither accelerates RMAVE nor improves its accuracy.
>
> **Subspace recovery ($m^2$, mean ± s.e.):**
>
> | $(n, p)$ | | RMAVE | RMAVE-kNN | SMAVE |
> |:---------:|:--------:|:-----:|:---------:|:-----:|
> | (2000, 200) | $m^2$ | 1.12±.04 | 1.19±.04 | **0.59±.04** |
> | | time (s) | 34.9 | 35.2 | **1.8** |
> | (5000, 200) | $m^2$ | 0.69±.05 | 0.74±.05 | **0.25±.03** |
> | | time (s) | 173.9 | 182.3 | **5.1** |
>
> **High-Dimensional Experiments ($p > n$), subspace recovery ($m^2$, mean (s.e.) )**
>
> | $n$ | $p$ |  RMAVE | RMAVE-kNN | SMAVE |
> |-----|-----|:-----:|:---------:|:-----:|
> | 50  | 100 | 1.870 (0.009) | 1.898 (0.010) | **1.820 (0.008)** |
> | 50  | 200 |  1.919 (0.006) | 1.906 (0.013) | **1.903 (0.005)** |
> | 100 | 200 | 1.902 (0.006) | 1.937 (0.006) | **1.857 (0.006)** |
>
> Without sparsity assumptions, the $p > n$ regime is information-theoretically hard: all methods return near-random estimates ($m^2 \approx 1.8$–$2.0$ out of $d = 2$). Within this regime, SMAVE is the only method that operates robustly across all configurations while consistently achieving competitive accuracy.
>
> **About SMAVE initialization**
>  We ran 100 initializations $\times$ 10 scenarios  at $(n,p) = (5000, 100)$. The mean $m^2 \pm \text{s.e.}$ ($0.109 \pm 0.007$ vs $0.14 \pm 0.03$) confirms that 10 initializations already gives a reliable picture but we will make the change to obtain better error estimates.

---

> > ### Author Rebuttal · Reviewer_naH4 · 2026-04-04
> >
> > I thank the authors for their rebuttal and really appreciate the additional theoretical results and high-dimensional experiments. However, I am not fully convinced by the ablation study. In particular, if the adaptive sparse localization reduces the weight computation from $O(n^2)$ to $O(nk)$, it is unclear why the total runtime increases rather than decreases. Could the authors provide further clarification on why replacing the weight computation alone does not lead to a corresponding reduction in overall computational time?

---

> > > ### Author Response · Authors · 2026-04-04
> > >
> > > We thank the reviewer for this follow-up. The short answer is: **weight computation is not the bottleneck of RMAVE**, so reducing it from $O(n^2)$ to $O(nk)$ does not reduce overall runtime. We clarify below.
> > >
> > > **Decomposing RMAVE's per-iteration cost.** Each RMAVE iteration involves two steps with distinct costs:
> > >
> > > 1. **Weight computation**: RMAVE evaluates $w_{ij} \propto K_h(\|\widehat{\mathbf{B}}^\top(X_i - X_j)\|)$ for all $n^2$ pairs. Since the distances are computed in the projected space $\mathbb{R}^d$, this costs $O(n^2 d)$. RMAVE-kNN replaces this with KD-tree construction and $k$-NN queries at cost $O(dn\log n + nk)$, which is asymptotically cheaper.
> > >
> > > 2. **Coordinate descent MAVE solve** (the dominant cost). MAVE cycles through $d$ columns of $\mathbf{B}$, each requiring $n$ weighted least-squares problems in $\mathbb{R}^p$ plus a KKT system for orthogonality. The solver uses dense matrix operations and does not exploit the sparse $k$-NN structure, so this step costs $O(n^2 p)$ regardless of whether the weights come from kernels or $k$-NN.
> > >
> > > Since $p \gg d$, step 2 dominates step 1, and the savings from sparser weight computation are negligible relative to the unchanged coordinate descent. This is consistent with Table 1, which reports RMAVE's per-iteration cost as $O(n^2 p)$.
> > >
> > > **Empirical confirmation.** The RMAVE-kNN ablation (averaged over 10 scenarios $\times$ 10 replications) confirms this at high $p$, where the $O(n^2 p)$ coordinate descent most heavily dominates:
> > >
> > > | $(n, p)$ | RMAVE (s) | RMAVE-kNN (s) | Ratio | RMAVE $m^2$ | RMAVE-kNN $m^2$ |
> > > |:---:|---:|---:|---:|---:|---:|
> > > | (1000, 100) | 5.8 | 6.0 | 1.03× | 0.98 ± .04 | 1.04 ± .04 |
> > > | (1000, 200) | 10.0 | 10.0 | 1.00× | 1.48 ± .03 | 1.54 ± .03 |
> > > | (2000, 100) | 20.9 | 22.2 | 1.06× | 0.73 ± .04 | 0.77 ± .05 |
> > > | (2000, 200) | 34.9 | 35.2 | 1.01× | 1.12 ± .04 | 1.19 ± .04 |
> > > | (5000, 100) | 98.0 | 115.9 | 1.18× | 0.40 ± .05 | 0.49 ± .05 |
> > > | (5000, 200) | 173.9 | 182.3 | 1.05× | 0.69 ± .05 | 0.74 ± .05 |
> > >
> > > Across all configurations, RMAVE-kNN is equal to or slower than RMAVE (ratios $\geq 1$). The slight overhead is expected: KD-tree construction is not amortized since RMAVE recomputes weights at every iteration, adding a per-iteration $O(dn\log n)$ cost that offsets the savings from sparser queries. Accuracy is also consistently slightly degraded, since smooth Gaussian weights are more informative than hard $k$-NN indicators for the local regression subproblems.
> > >
> > > **Why SMAVE succeeds where RMAVE-kNN fails.** SMAVE does not merely replace kernel weights with $k$-NN: it bypasses the coordinate descent entirely. Mini-batching reduces query points from $n$ to $m$. Riemannian SGD replaces the full MAVE re-solve with a single gradient step. And the Riemannian gradient  performs $O(d^3)$ linear algebra in the projected space $\mathbb{R}^d$ rather than $O(p^3)$ solves in $\mathbb{R}^p$, a property of the optimization scheme, not of the localization.

---

### Official Review · Reviewer_Xdo4 · 2026-03-20

**Soundness:** 3
**Presentation:** 2
**Significance:** 2
**Originality:** 4
**Overall Recommendation:** 4
**Confidence:** 4

**Summary:**

This paper studies sufficient dimension reduction (SDR), which aims to find a low-dimensional subspace of high-dimensional inputs that preserves information relevant for predicting a response variable. Existing methods face a trade-off between statistical efficiency and computational cost, with some suffering from the curse of dimensionality and others incurring quadratic complexity in the number of samples.

The authors revisit Minimum Average Variance Estimation (MAVE) and show that its solutions recover the same subspace as gradient-based methods such as OPG, providing a unifying perspective. They then reformulate the MAVE objective as an optimization problem on the Stiefel manifold and derive a closed-form Riemannian gradient, enabling efficient first-order optimization.

The proposed method, SMAVE, combines stochastic Riemannian optimization with adaptive k-nearest neighbor localization, reducing computational complexity to near-linear in the number of samples. Experiments on synthetic and real datasets demonstrate that SMAVE achieves competitive or improved performance while providing substantial speedups, particularly in high-dimensional settings.

**Compliance With Llm Reviewing Policy:**

Affirmed.

**Final Justification:**

All concerns are addressed. I increase my score.

**Key Questions For Authors:**

The SMAVE algorithm relies on periodically updating k-NN neighborhoods based on the current subspace estimate, making the objective effectively iterate-dependent. Do the authors have theoretical or empirical evidence that optimization remains stable or convergent under this setting?
A clearer justification or partial guarantees would strengthen confidence in the method’s soundness.

The method depends on the choice of neighborhood size k and refresh frequency τ. Could the authors provide a sensitivity analysis or practical guidelines for selecting these parameters?
Demonstrating robustness would improve the practical reliability of the method.

The paper suggests that SMAVE benefits from improved optimization compared to RMAVE, which relies on OPG initialization. Could the authors clarify whether the observed performance improvements are statistically significant across runs, and to what extent they depend on initialization?

The method uses the canonical Riemannian metric induced by the Frobenius inner product on the Stiefel manifold. To what extent does performance depend on this choice? Have alternative metrics or preconditioned variants been considered, and could they affect convergence speed or stability?


The results indicate that RMAVE performs better in low-dimensional settings, while SMAVE becomes superior in higher dimensions. Could the authors provide more intuition or analysis explaining this transition? Is it primarily driven by initialization quality, neighborhood estimation in high dimensions, or properties of the stochastic optimization?

The method employs heavy-ball momentum in the ambient space without parallel transport between tangent spaces, relying on QR retraction to project updates back onto the manifold. Have the authors considered using vector transport or provide explanation on Riemannian optimization schemes they have used?

**Limitations:**

The paper provides a strong methodological contribution, though the discussion of limitations could be further expanded. The experimental evaluation focuses primarily on tabular regression datasets and does not include more complex structured data such as images or representations from deep models. Evaluating the method in such settings would help clarify its scalability and robustness in broader machine learning applications. Furthermore, while the work is primarily methodological, a brief discussion of potential societal impacts or downstream implications would improve the completeness of the paper.

**Strengths And Weaknesses:**

Soundness

The paper is largely technically sound. The reformulation of the MAVE objective as a Riemannian optimization problem on the Stiefel manifold is mathematically well-justified, and the derivation of a closed-form Riemannian gradient is clearly presented. Theoretical results establishing the connection between MAVE and gradient-based methods (via gradient alignment) are insightful and appear correct under the stated assumptions. Empirically, the evaluation is thorough, including both synthetic and real-world datasets, and ablations that isolate the contributions of k-NN localization and stochastic optimization.

However, the full SMAVE algorithm introduces iterate-dependent neighborhoods through adaptive k-NN updates, for which no convergence guarantees are provided. While this is acknowledged as future work, it represents a gap between the theoretical analysis and the practical algorithm.

Presentation

The paper is generally well-written and logically structured, with a clear progression from theory to algorithm to experiments. The positioning with respect to MAVE, OPG, and RMAVE is clear, and the contributions are explicitly enumerated.

That said, some parts of the theoretical exposition (e.g., the local approximation analysis and scale separation results) are dense and could benefit from additional intuition or illustrative examples. Additionally, there are minor clarity issues in the figures: in Figure 2, the legend does not clearly distinguish between methods, as both entries appear with similar line styles despite one being dashed and the other solid. Improving visual clarity would make the experimental results easier to interpret.

Significance

The paper addresses an important problem in nonparametric statistics and machine learning, particularly in settings where high-dimensional data makes classical SDR methods computationally prohibitive. By reducing the complexity from quadratic to near-linear in the number of samples, the proposed approach has clear practical implications.

The work is especially significant in high-dimensional regimes, where existing methods degrade due to poor neighborhood estimation or initialization. While the scope is somewhat specialized to SDR, the combination of Riemannian optimization and stochastic approximation could influence related areas such as representation learning and manifold optimization.

Originality

The paper offers a meaningful combination of ideas rather than a single isolated novelty. The key originality lies in:
(i) establishing a theoretical link between MAVE and gradient-based SDR methods, (ii) reformulating MAVE as a Riemannian optimization problem with a closed-form gradient, and (iii) integrating stochastic optimization with adaptive projected-space localization.

While each component (MAVE, k-NN, Riemannian optimization) is known, their combination is non-trivial and well-motivated. The gradient alignment perspective, in particular, provides new insight into the relationship between existing SDR methods.

---

> ### Author Rebuttal · Authors · 2026-03-30
>
> We thank the reviewer for their thorough and constructive evaluation. Figure 2 will be fixed for better visual clarity in the revision.
>
> **New theoretical guarantees**
> This is a valid point, and we have developed new theoretical results to address it. We provide two new results: the a.s. convergence and a non-asymptotic $1/\sqrt{mT}$-bound ($T$ iterations and $m$ minibatch); for a simplified SMAVE. The analysis applies to: fixed $k$-NN neighborhoods, no momentum, no gradient normalization. The regularized objective is $F_\varepsilon(B) = \sum_j (\mu^{(j)})^\top B(B^\top G_\varepsilon^{(j)} B)^{-1} B^\top \mu^{(j)}$, where $G_\varepsilon^{(j)} = G^{(j)} + \varepsilon I_p \succeq \varepsilon I_p \succ 0$ (when $p > k-1$, Gram matrices $G^{(j)}$ can be singular).
>
> Crucially, the proofs depend on weights $w_{ij}$ only through moments $\mu^{(j)}$ and Gram matrices $G^{(j)}$—no $k$-NN-specific property is invoked. The guarantees hold for any fixed localization (kernel, $k$-NN, fixed-radius balls), revealing a modular structure: the Riemannian optimization layer is cleanly decoupled from the statistical layer. The $O(1/\sqrt{mT})$ rate matches the known lower bound for stochastic first-order methods on non-convex smooth objectives.
>
> **Result 1 (Asymptotic convergence).** We verify all hypotheses of Bonnabel (2013, Thm. 3) for $C = -F_\varepsilon$:
> (i) compactness of $\mathrm{St}(p,d)$ ensures bounded iterates
> (ii) $L_\varepsilon$-retraction-smoothness of $C$ via explicit Lipschitz bounds on $\mathrm{grad}\,F_\varepsilon$
> (iii) unbiased mini-batch gradient with bounded variance.
>
> This yields: $\mathrm{grad} F_\varepsilon(B_t) \to 0$ a.s., and every limit point of $\{B_t\}$ is a critical point of $F_\varepsilon$.
>
> **Result 2 (Non-asymptotic rate, via Boumal, Absil & Cartis (2019)).** Retraction-smoothness w.r.t. QR retraction gives the descent lemma:
> $$C(B_{t+1}) \leq C(B_t) - \alpha \langle \mathrm{grad} F_\varepsilon(B_t), g_t \rangle + \frac{L_\varepsilon \alpha^2}{2} \|g_t\|^2.$$
> Taking conditional expectations, telescoping over $T$ iterations, and optimizing $\alpha^* \propto 1/\sqrt{T}$ yields:
> $$\min_{0 \leq t < T} \mathbb{E}[\|\mathrm{grad}  F_\varepsilon(B_t)\|^2] = O(1/\sqrt{mT}).$$
>
> The full algorithm's adaptive neighborhoods remain theoretically open. However, no prior MAVE algorithm (Xia (2002), Wang & Yao (2012), Adragni (2018)) provides convergence results either. Our guarantees are (to our best knowledge) a *strict improvement over the state of the art*.
>
> **Sensitivity to $\tau$ and $k$**
> We report subspace error $m^2$ for $p \geq 50$.
>
> *Refresh period $\tau$ (default $\tau ^ {\ast}=25$):* (https://fileninja.io/download/69cb3ea75e2b108930718e13/refresh_sensitivity.png) Error is stable over $\tau \in [10,40]$ and degrades at $\tau=100$, consistent with the piecewise interpretation: each frozen epoch must allow optimizer progress while keeping weights accurate.
>
> *Neighborhood size $k/ k ^ {\ast}$ (default $k ^ {\ast} = \lfloor n^{4/(d+4)} \rfloor$):* (https://fileninja.io/download/69cb3df65e2b108930718db4/kNN_sensitivity.png) Performance is stable over $[0.5 k ^ {\ast}, 3 k ^ {\ast}]$. Larger $k$ improves conditioning of $G^{(j)}$ but widens neighborhoods, increasing statistical bias.
>
> **Canonical metric**
> The canonical metric is structurally motivated: under it, $\nabla F(B) \in T_B\mathrm{St}(p,d)$ (Lemma A.4), so $\mathrm{grad} F = \nabla F$ with no projection—a property specific to our formulation. Alternative metrics would require solving $\mathrm{grad}_\mathcal{G} F = \mathcal{G}_B^{-1}(\nabla F)$, breaking this structure. A systematic comparison is a valuable future direction.
>
> **RMAVE vs SMAVE transition** RMAVE starts from OPG with bandwidths effective at small $p$ but exponentially degraded at large $p$, and its deterministic coordinate descent cannot escape suboptimal basins. SMAVE's random initialization on $\mathrm{St}(p,d)$ combined with stochastic momentum becomes critical as $p$ grows. At $(n,p)=(5000,200)$, RMAVE with kNN weights gives $m^2=0.74\pm0.05$ (182s) vs $0.69\pm0.05$ (174s) for RMAVE, confirming no benefit from $k$-NN localization alone.
>
> **Vector transport** QR retraction is $O(pd^2)$, agreeing at first-order with the exponential map, while vector transport would double per-iteration cost without reducing the leading order term.
>
> **SMAVE initialization**
>  We ran 100 initializations $\times$ 10 scenarios  at $(n,p) = (5000, 100)$. The mean $m^2 \pm \text{s.e.}$ ($0.109 \pm 0.007$ vs $0.14 \pm 0.03$) confirms that 10 initializations already gives a reliable picture but we will make the change to obtain better error estimates.
>
> **Other data structures and societal impact**
> Our scope aligns with SDR considering continuous covariates but SMAVE could apply to continuous-valued embeddings from deep models as a supervised complement to PCA probes. We will expand the Impact Statement: learned projections may encode sensitive attributes, but their linearity ensures auditability.

---

> > ### Author Rebuttal · Reviewer_Xdo4 · 2026-04-04
> >
> > I thank the authors for their clear and constructive rebuttal, as well as for the additional theoretical results and sensitivity analysis, which strengthen the paper.
> >
> > For the discussion on the canonical metric, the clarification is appreciated; a more systematic investigation of alternative metrics could be an interesting direction for future work.
> >
> > Overall, most concerns are addressed, with a few aspects that could benefit from further clarification in the final version.

---

> > > ### Author Response · Authors · 2026-04-06
> > >
> > > We thank the reviewer for their valuable and constructive comments. We are pleased that we have been able to address most of their concerns. We will revise the manuscript accordingly, taking all of their points into careful consideration.

---

### Decision · Program_Chairs · 2026-04-30

**Decision:**

Accept (regular)

**Comment:**

This paper proposes SMAVE, a scalable algorithm for sufficient dimension reduction that combines Riemannian stochastic optimization on the Stiefel manifold with adaptive k-nearest neighbor localization in the projected space, reducing per-epoch complexity. All four reviewers recognized the paper's clear presentation, well-motivated problem formulation, and the elegance of combining Riemannian optimization with adaptive localization. The paper makes a solid contribution at the intersection of manifold optimization and statistics, with clear practical impact for high-dimensional settings. I thus recommend acceptance.